# Programmable RNA detection with CRISPR-Cas12a

**Santosh R. Rananaware[1], Emma K. Vesco [1,5], Grace M. Shoemaker [1], Swapnil S. Anekar[1], Luke Samuel W. Sandoval [2], Katelyn S. Meister [1], Nicolas C. Macaluso[1], Long T. Nguyen[1] & Piyush K. Jain [1,3,4]** ✉

Cas12a, a CRISPR-associated protein complex, has an inherent ability to cleave DNA substrates and is utilized in diagnostic tools to identify DNA molecules. We demonstrate that multiple orthologs of Cas12a activate trans-cleavage in the presence of split activators. Specifically, the PAM-distal region of the crRNA recognizes RNA targets provided that the PAM-proximal seed region has a DNA target. Our method, Split Activator for Highly Accessible RNA Analysis (SAHARA), detects picomolar concentrations of RNA without sample amplification, reverse-transcription, or strand-displacement by simply supplying a short DNA sequence complementary to the seed region. Beyond RNA detection, SAHARA outperforms wild-type CRISPR-Cas12a in specificity towards point-mutations and can detect multiple RNA and DNA targets in pooled crRNA/Cas12a arrays via distinct PAM-proximal seed DNAs. In conclusion, SAHARA is a simple, yet powerful nucleic acid detection platform based on Cas12a that can be applied in a multiplexed fashion and potentially be expanded to other CRISPR-Cas enzymes.

CRISPR (Clustered Regularly Interspaced Short Palindromic Repeats) is an adaptive immune system encoded within prokaryotes that has evolved to counter invasion by foreign nucleic acids from bacteriophages and plasmids[1,2]. Upon infection, the invading DNA sequences are captured and integrated into the host genome between an array of repeat sequences. The captured DNA sequences are called 'spacers' and they provide a genetic memory of prior infections[3]. For prokaryotic immunity, the CRISPR locus is transcribed and processed to generate multiple mature CRISPR RNAs (crRNA), each encoding a unique spacer sequence. The crRNA forms a complex with an RNA-guided endonuclease, Cas (CRISPR-associated) protein. In the presence of nucleic acids complementary to the crRNA sequence, the CRISPR/Cas system enables the cleavage of nucleic acids. There are several diverse naturally occurring CRISPR/Cas systems found in prokaryotes[4,5]. Among these, Cas12a is a class 2, type V RNA-guided DNA endonuclease[6]. Since its discovery, it has been widely used for genome editing as well as molecular diagnostic applications[7–11].

Structural and biochemical studies have shown that Cas12a can catalyze the cleavage of DNA substrates[12–14] but there are no reports of targeted RNA cleavage by Cas12a, without pre-processing steps such as reverse transcription. Recently there has been a report of a novel enzyme named Cas12a2 that sometimes co-occurs with Cas12a systems in bacteria and can use the Cas12a crRNA, but recognizes an RNA target instead of a double-stranded DNA[15]. A special characteristic of the type V Cas12 family of enzymes is their ability to initiate rapid and indiscriminate cleavage of any non-specific single-stranded DNA (ssDNA) molecules in their vicinity after target-specific recognition and cleavage[16,17]. This unique catalytic property, known as *trans*-cleavage has been harnessed to engineer CRISPR-based diagnostic tools that rely on the cleavage of FRET reporters upon target recognition[18].

So far, Cas12a-based tools have been limited to the detection of DNA substrates, unless they are coupled with additional steps involving reverse transcription or strand displacement[19–21]. Previously we have shown that CRISPR-Cas12a can tolerate DNA/RNA

[1]Department of Chemical Engineering, University of Florida, Gainesville, FL, USA. [2]Department of Biology, CLAS, University of Florida, Gainesville, FL, USA. [3]Department of Molecular Genetics and Microbiology, University of Florida, Gainesville, FL, USA. [4]UF Health Cancer Center, University of Florida, Gainesville, FL, USA. [5]Present address: Department of Molecular and Human Genetics, Baylor College of Medicine, Houston, TX, USA. ✉e-mail: jainp@ufl.edu

heteroduplexes, but only when RNA is located at the non-target strand[22]. This can be used to detect RNA targets with Cas12a by simply creating a heteroduplex using a reverse transcription step without amplification[22,23]. A reverse transcription step is inconvenient because it adds to the time, cost, error, and complexity of the assay. The other alternative is to use an RNA-targeting enzyme such as Cas12a2[15], Cas12g[24], or Cas13a-d[25–27]; however, these systems can only detect RNA.

Cas12a orthologs are known to require a short protospacer adjacent motif (PAM) to be present on the target DNA to initiate recognition and cleavage[6]. It has been previously shown that DNA-cleaving enzymes such as Cas9 can be manipulated to also cleave RNA through the addition of a PAMmer sequence, a PAM-containing oligonucleotide, which is annealed to the target ssRNA to initiate cleavage[28]. However, similar approaches have not yet been investigated with *trans*-cleaving Cas enzymes like Cas12, primarily because the PAM recognition mechanism is different between Cas9 and Cas12 enzymes[29]. For instance, unlike Cas9, Cas12a can recognize and even trigger *trans*-cleavage activity using a PAM-less ssDNA. To date, there is no single CRISPR-Cas system identified that can innately tolerate both DNA and RNA substrates to trigger *trans*-cleavage.

In this work, we show that Cas12a can tolerate RNA substrates at the PAM-distal end of the crRNA and initiate *trans*-cleavage activity. While the PAM-proximal seed region of the crRNA strictly tolerates DNA substrates, the PAM-distal end of the crRNA can tolerate both RNA and DNA substrates with multiple Cas12a orthologs. Thus, by merely supplying a short ssDNA or a PAM-containing dsDNA at the seed region of the crRNA we can detect RNA substrates at the 3′-end of the crRNA. This specific property is the basis of our method for RNA detection using Cas12a named <u>S</u>plit <u>A</u>ctivators for <u>H</u>ighly <u>A</u>ccessible <u>R</u>NA <u>A</u>nalysis (SAHARA).

SAHARA achieves reverse transcription (RTx)-free detection of picomolar levels of DNA as well as RNA without amplification and can detect clinically relevant targets including hepatitis C virus (HCV) RNA and microRNA-155 (miR-155). We show that SAHARA works robustly at room temperature and has improved specificity for point mutation detection compared to conventional CRISPR/Cas12a systems. We demonstrate that its activity requires the seed region DNA which allows for multiplexed assays. We use this switch capability to simultaneously detect different DNA and RNA targets. We also show that SAHARA works with Cas13b to perform multiplexed detection of different RNA targets. These key findings provide insight into the substrate requirements for the *trans*-cleavage activity of Cas12a, which are essential for SAHARA, a valuable and versatile tool that can simultaneously detect both DNA and RNA substrates.

## Results

### Cas12a orthologs tolerate split ssDNA activators for *trans*-cleavage activity

LbCas12a, AsCas12a, and ErCas12a are orthologs of Cas12a nucleases that are derived from *Lachnospiraceae bacterium* ND2006, *Acidaminococcus* sp. BVL36, and *Eubacterium rectale* and are simply referred to here as Lb, As, and Er, respectively[30–33] (Fig. S1). The mature crRNAs for each ortholog are 41–44 nt in length, containing ~19–21 nt of scaffold and the remaining ~20-24 nt of spacer[6].

We wondered whether two different ssDNA targets, each binding to a different position of the same crRNA, can be used to initiate the *trans*-cleavage activity of Cas12a (Fig. 1a, S2). To test the minimum length of activators tolerated by Cas12a, we designed several short ssDNA target activators of lengths ranging from 6-20 nt that were complementary to either the PAM-proximal (Pp) seed region or the PAM-distal (Pd) end of the crRNA (Fig. 1b–d, S3).

We first performed in vitro *trans*-cleavage assays with individual truncated activators using three different orthologs of Cas12a. We found that the *trans*-cleavage activity is sensitive to truncations of the ssDNA activators across the tested orthologs (Fig. 1b–d). Compared to

the full-length 20-nt activator, the *trans*-cleavage activity for a 16-nt activator diminishes by as much as 50-70-fold and is completely lost for activators less than or equal to 14-nt in length. Unlike LbCas12a and ErCas12a which show no *trans*-cleavage, AsCas12a shows a detectable activity for the target of length 14-nt. These results corroborate an earlier study that demonstrated a crRNA-target DNA interaction longer than 14-nt is necessary to initiate the indiscriminate *trans*-cleavage activity of Cas12a[34].

Next, we tested to check if the simultaneous addition of two truncated activators, each binding to different regions of the crRNA and together mimicking a full-length target, would be able to regain the diminished *trans*-cleavage activity observed for shorter (<=14-nt) activators (Fig. S2). For this, we analyzed the activity produced by truncated activators shorter than 14-nt in a combinatorial fashion (Fig. 1e–g). We observed that while the individual truncated activators failed to trigger any *trans*-cleavage activity, a split-activator combination was able to partially regain the diminished activity when the combined lengths were greater than or equal to 20-nt. This demonstrated that Cas12a enzymes have the ability to accept two different activators in a split-activator fashion and still initiate *trans*-cleavage.

### Cas12a tolerates RNA activators at the PAM-distal 3′-end of the crRNA

After observing the *trans*-cleavage behavior of Cas12a orthologs towards truncated ssDNA split-activators, we wondered how sensitive the Pd and Pp regions were to other nucleic acid substrates such as RNA and dsDNA. We were particularly curious to see how RNA substrates would be tolerated in the split-activator system since Cas12a is not known for initiating *trans*-cleavage after targeting RNA. To study this, we designed 10-nt ssDNA, dsDNA, and RNA activators complementary to either the Pp or the Pd region of the crRNA (Fig. 2a).

We tested the detection of different ssDNA, dsDNA, and RNA activators in a combinatorial fashion (Fig. 2b–d). Upon switching from ssDNA to RNA at the Pp seed region, we observed that the *trans*-cleavage activity completely vanishes for Lb and Er irrespective of the type of substrate supplied at the Pd. This demonstrated a strict DNA substrate requirement at the seed region. On the contrary, AsCas12a seemed to tolerate even RNA substrates at the Pp region, hinting at distinct structural recognition features from other Cas12a orthologs. Interestingly, all three orthologs were observed to tolerate RNA substrates at the Pd region of the crRNA, provided a short piece of DNA substrate is supplied at the Pp region. The DNA binding at the Pp can be ssDNA or dsDNA. However, we observed that the *trans*-cleavage activity is significantly boosted for As and Er with a PAM-containing dsDNA at the Pp region instead of an ssDNA.

Interested in the ability of AsCas12a to tolerate RNA sequences at the Pp region, we wondered if there were other orthologs capable of the same. To investigate the tolerance of Pp RNA detection with other Cas12a, we tested three orthologs—*Helcococcus kunzii* ATCC 51366 (Hk), *Moraxella bovoculi* 237 (Mb), and *Butyrivibrio sp.*NC3005 (Bs) (Fig. S4). We chose these specific orthologs because we have characterized their trans-cleavage activity in an earlier study[35]. Furthermore, we had previously determined that Hk is phylogenetically closely related to As, so we were curious to see if it would show similar tolerance towards Pp RNA. Upon testing the new orthologs with a 10-nt Pp RNA or equivalent Pp DNA (with a 10-nt Pd DNA in each case), we observed that all three orthologs displayed trans-cleavage activity only with Pp DNA but not Pp RNA. Suggesting that, unlike AsCas12a, these Cas12a also have a strict preference for DNA only at the PAM-proximal region.

While a fully complementary RNA activator is not recognized by Cas12a, we have shown that RNA can be recognized at the 3′-end of the crRNA if DNA is bound at the seed region. Therefore, somewhere between the seed region and the 3′ spacer-end of the crRNA, RNA recognition starts being tolerated. To examine the exact position at

which RNA detection begins, we designed increasingly long RNA activators of length ranging from 6-14 nt that are complementary to the Pd region of the crRNA, and their corresponding DNA activators of different lengths binding to the Pp region (Fig. S8). We tested these DNA-RNA combinations in tandem and observed that while RNA activators of length 6-10 nt are recognized by Cas12a, increasing the length of the RNA activator beyond 10-nt stops any trans-cleavage activity. This suggests that the Pp 10-nt nucleotides of the crRNA spacer can exclusively recognize DNA targets while the Pd 10-nt can detect both RNA and DNA.

Previously, we have shown that adding a 7-nt DNA extension at the 3′-end of the crRNA significantly boosts the trans-cleavage activity of LbCas12a[22]. We were curious to check if these modified crRNAs (termed ENHANCE or EN) would have any significant effect on the *trans*-cleavage activity produced by the split-activator

system. We analyzed the detection of ssDNA and ssRNA substrates using the split-activator system with both the wild-type (WT) and EN crRNAs. We observed that both ssDNA and ssRNA had significantly greater activity with the EN crRNA for LbCas12a, while AsCas12a and ErCas12a were unaffected by the different crRNA designs. This is consistent with previous observations with ENHANCE[22]. We observed ErCas12a to have the highest activity with the split-activator system irrespective of the crRNA design being used (Fig. 2f–h).

Chimeric DNA-RNA hybrid guides have been previously used to increase the sensitivity and reduce the off-target effects for both Cas9 and Cas12 nucleases[22,36,37]. We questioned if the use of a chimeric DNA-RNA hybrid crRNA would help with the detection of split-activator sequences. To investigate this, we designed two chimeric crRNAs by changing either 12-nt at the Pp region of the crRNA to DNA

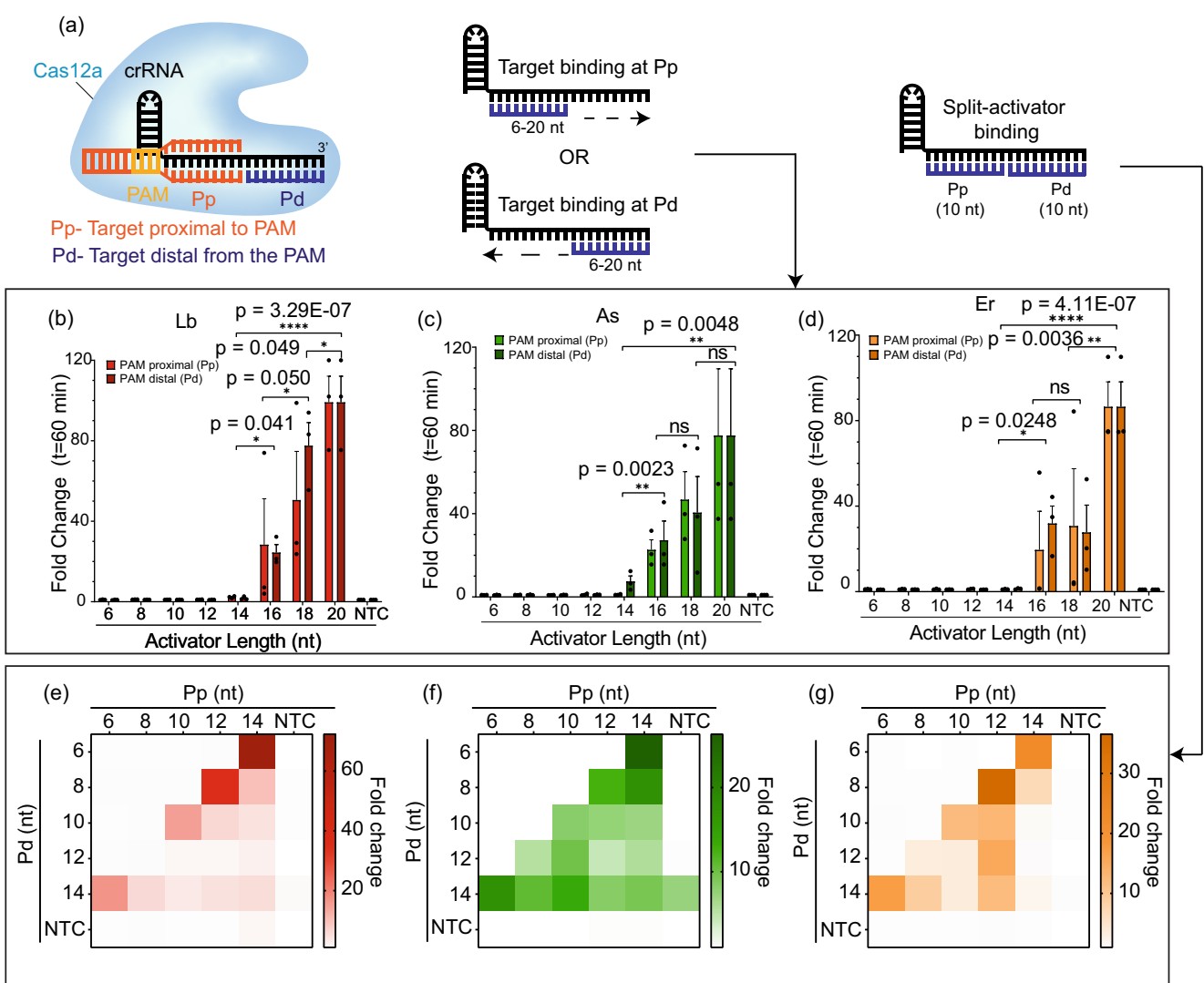

**Fig. 1 | Cas12a orthologs tolerate short ssDNA activators (6-12 nt) when added in combination. a** Schematic representation of a crRNA-Cas12a complex performing *trans*-cleavage of ssDNA reporters following the recognition of two split-activators. **b**–**d** Fold change of fluorescence intensity normalized to No Target Control (NTC) at *t* = 60 minutes of in vitro *trans*-cleavage assay with Cas12a orthologs (red = LbCas12a, green = AsCas12a, orange = ErCas12a) activated by individual truncated ssDNA activators of length 6–20 nt. Statistical comparisons are made for a combination of both Pp and Pd data sets between the different lengths of the targets. Statistical analysis for *n* = 3 biologically independent replicates was performed using a two-tailed *t*-test where ns = not

significant with *p* > 0.05, and the asterisks (\**p* ≤ 0.05, \*\**p* ≤ 0.01, \*\*\**p* ≤ 0.001, and \*\*\*\**p* ≤ 0.0001) denote significant differences. Error bars represent Mean value +/− Standard Error of Mean (SEM) (*n* = 3). **e**–**g** Heat maps representing fold change with respect to NTC at *t* = 60 min of an in vitro *trans*-cleavage assay activated by combinations of truncated ssDNA activators of different lengths ranging from 6 to 14 nt in the Pp and Pd regions. The reactions contained 25 nM truncated ssDNA GFP-activators, 60 nM Cas12a, and 120 nM crGFP and were incubated for 60 min at 37 °C. The NTC represents the condition when neither the Pp nor the Pd activator is present in the reaction. Source data are provided as a Source Data file.

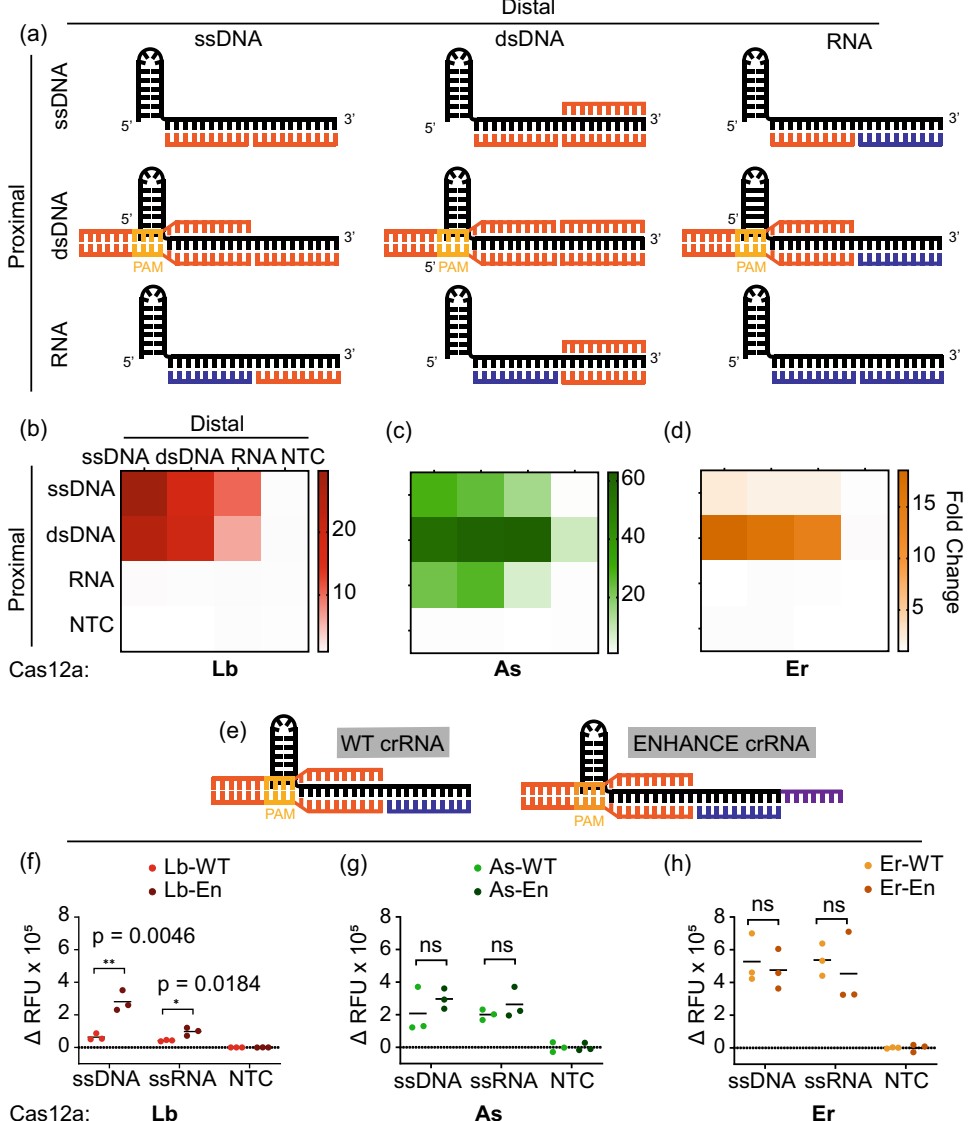

**Fig. 2 | Split-activator detection of ssDNA, dsDNA, and RNA substrates by Cas12a. a** Schematic representation of Cas12a activated by combinations of ssDNA/dsDNA (orange), and RNA (blue) in the PAM proximal and PAM distal regions. **b–d** Heat maps representing the fold changes in the fluorescence intensity of in vitro *trans*-cleavage assay (n = 3) with Cas12a orthologs for the combinatorial schemes seen in (**a**). **e–h** Comparison of the WT crRNA and ENHANCE crRNA for in vitro *trans*-cleavage assay of split activators. The line represents the mean of the scatter plot in each group. Statistical analysis for n = 3 biologically independent replicates comparing WT vs. En data in each group was performed using a two-tailed *t*-test where ns = not significant with $p > 0.05$, and the asterisks (*$p \leq 0.05$, **$p \leq 0.01$, ***$p \leq 0.001$, and ****$p \leq 0.0001$) denote significant differences. Note that ssDNA and ssRNA substrates were used as targets in the Pd region while dsDNA was supplied in the Pp. Reactions were incubated for 60 min at 37 °C. The reactions contained 25 nM of each truncated activator, 60 nM Cas12a, and 120 nM crRNA. Source data are provided as a Source Data file.

(crRNA-12D8R) or 8-nt of the Pd region of the crRNA to DNA (crRNA-12R8D) (Fig. S5). However, our results indicated that the chimeric DNA-RNA hybrid guides performed poorly in the detection of split-activators (Fig. S6).

### Development of SAHARA for the detection of different lengths of RNA targets

Using our observations, we developed SAHARA (Split Activators for Highly Accessible RNA Analysis) a tool for direct RNA detection with Cas12a. The design of SAHARA was such that a short (12-nt), synthetic PAM-containing dsDNA activator (named 'S12' due to its length) would bind to the Pp-end of the crRNA, while any target RNA sequence can be detected at the Pd-end. While it was clear that short RNA activators could be easily detected by SAHARA, this study aimed to aid in the optimization of detection for a variety of targets

and lengths. Therefore, we designed two RNA targets: a short RNA (20-nt) and a longer RNA (~730-nt) containing the same 20-nt target sequence (Fig. 3a).

To test these targets, we designed two crRNAs—a canonical WT as well as a SAHARA crRNA that only binds to the target at the Pd end while binding to a synthetic S12 at the Pp end. We observed that while the WT CRISPR-Cas system failed, SAHARA was robustly able to detect RNA activators of different lengths (Fig. 3b–d). ErCas12a showed the best detection for both short- and long-RNA targets, while AsCas12a performed the worst. LbCas12a had decent activity for the short RNA target, but it performed significantly worse as the length of the RNA increased (Fig. 3a–d). We also observed that SAHARA does not show any RNA detection if a non-targeting 'scrambled' S12-activator is used instead of a canonical on-target S12, confirming S12-dependent activation of Cas12a activity (Fig. 3e–g).

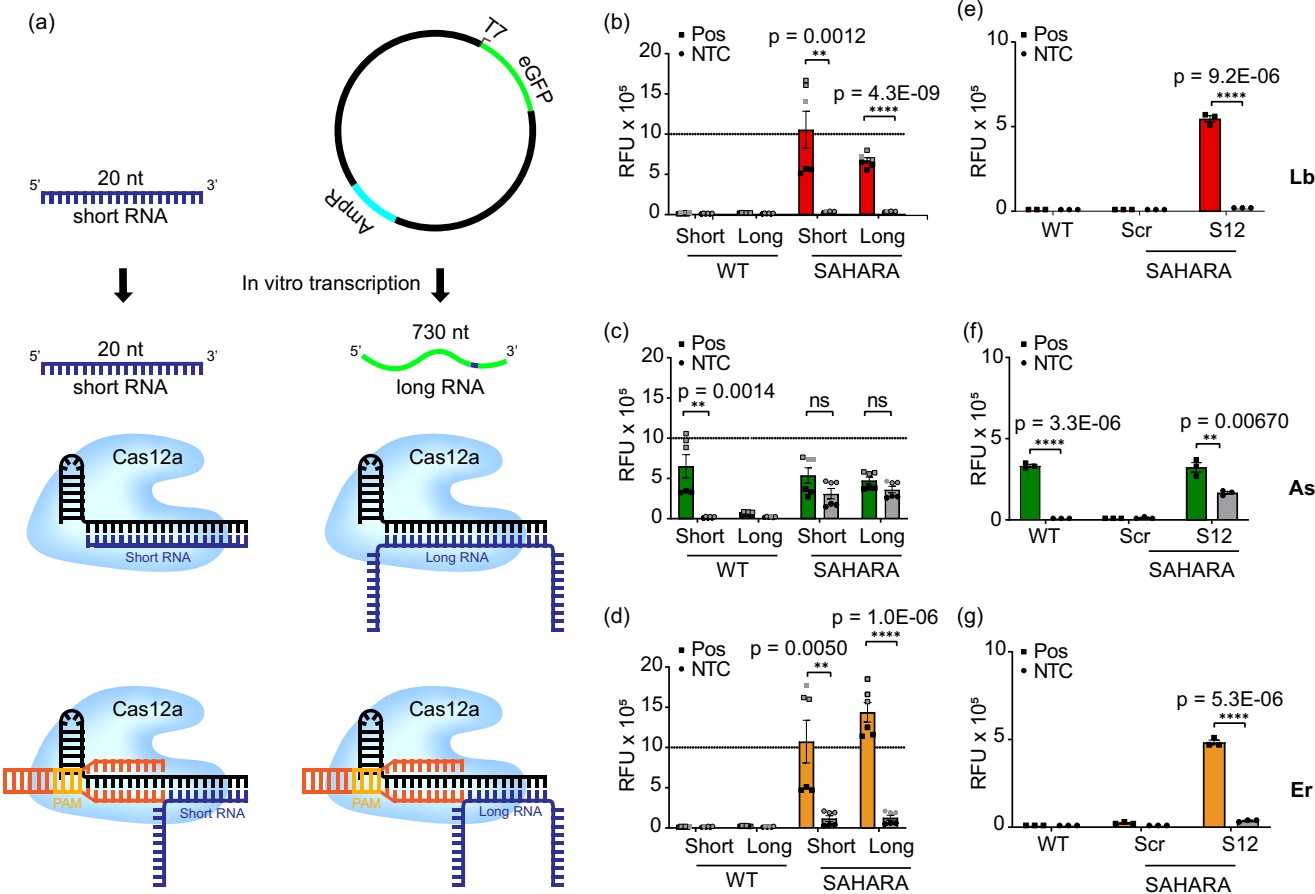

**Fig. 3 | Development of SAHARA for the detection of a wide range of RNA targets. a** Schematic representation of Cas12 complexed with WT vs. SAHARA crRNA and activated by either a short (20-nt) or a long (730-nt) RNA activator. **b**–**d.** Comparison of *trans*-cleavage activity among Cas12a orthologs for the short vs. long combinatorial schemes seen in (**a**). The plot represents raw fluorescence units (RFU) plotted for time $t = 60$ min. For this experiment, two biological replicates each with three technical replicates were performed. The data points for the two biological replicates are represented by black and gray, respectively. **e**–**g** Detection

of the long RNA target with SAHARA by using either a non-targeting scrambled S12 (SR-Scr) or a targeting S12 (SR-S12). The plot represents RFU at $t = 60$ min. Error bars for all charts represent mean value +/− SEM. Statistical analysis for $n = 6$ samples comparing short vs. long targets and $n = 3$ replicates comparing Pos vs. NTC was performed using a two-tailed *t*-test where ns = not significant with $p > 0.05$, and the asterisks (*$P \le 0.05$, **$P \le 0.01$, ***$P \le 0.001$, ****$P \le 0.0001$) denote significant differences. Source data are provided as a Source Data file.

Next, we studied whether the length of the crRNA has any effect on the RNA detection capability of SAHARA. We hypothesized that longer crRNAs would be better for RNA detection with SAHARA on account of their higher specificity of detection due to the increased number of base pairs between the activator and the target. To test this, we designed crRNAs of length ranging from 24–28 nt. Each crRNA was complementary to their respective S12 activators at their Pp end while binding to the increasingly longer length of the target RNA at the Pd (Fig. S7). While all three crRNAs tested displayed some level of RNA detection, the 24-nt crRNA had the best activity for RNA detection with SAHARA.

While the catalytic activity of Cas12a enzymes is optimal at 37 °C, the short (12-nt) DNA activators that are necessary for SAHARA to function, can bind in a more stable manner at temperatures lower than 37 °C, due to their lower Tm. To study this, we tested the detection of an ssDNA and ssRNA target each at 37 °C and room temperature (RT) (Fig. S9). We observed that both targets can be detected at RT. However, the activity drops by 3- to 5-fold depending on the Cas12a ortholog being used.

Cas12a effectors are metal-dependent endonucleases and therefore, the type and concentration of metal ions used in the reaction can have a significant effect on their activity. While $Mg^{2+}$ ions are routinely used in Cas12a-based applications, $Mn^{2+}$ ions have also been shown to

work well[38]. To study the effect of different ions on the activity of SAHARA, we tested a range of different cations ($NH_4^+$, $Rb^+$, $Mg^{2+}$, $Zn^{2+}$, $Cu^{2+}$, $Co^{2+}$, $Ca^{2+}$, $Ni^{2+}$, $Mn^{2+}$, and $Al^{3+}$) and discovered that most cations severely inhibited the trans-cleavage activity of SAHARA (Fig. S10).

Among the different divalent metal ions tested, $Mg^{2+}$ ions displayed the highest activity in-vitro, consistent with the literature[22]. Therefore, we characterized the effect of increasing the concentration of $Mg^{2+}$ ions on the activity. By varying the amount of $Mg^{2+}$ ions in the Cas12a reaction, we observed a significant increase in the activity with increasing concentration up to 15 mM. Increasing the $Mg^{2+}$ concentration beyond 15 mM slightly decreases the activity, suggesting 15 mM is the optimal concentration of $Mg^{2+}$ for SAHARA (Fig. S11).

Finally, the pH of the reaction can have a notable effect on the charge of the enzyme and can subsequently affect its activity. We screened the activity of SAHARA in different buffers with pH ranging from 5.5 to 9.25. We observed that SAHARA has optimal activity at pH = 7.9 and the activity drops sharply upon further increasing or decreasing the pH (Fig. S12). There was no observed activity at pH = 9.25 for any of the three orthologs tested.

**Detection of HCV and miRNA-155 with SAHARA**
To validate SAHARA with clinically relevant RNA targets, we designed multiple crRNAs targeting HCV RNA and miR-155. HCV is a viral

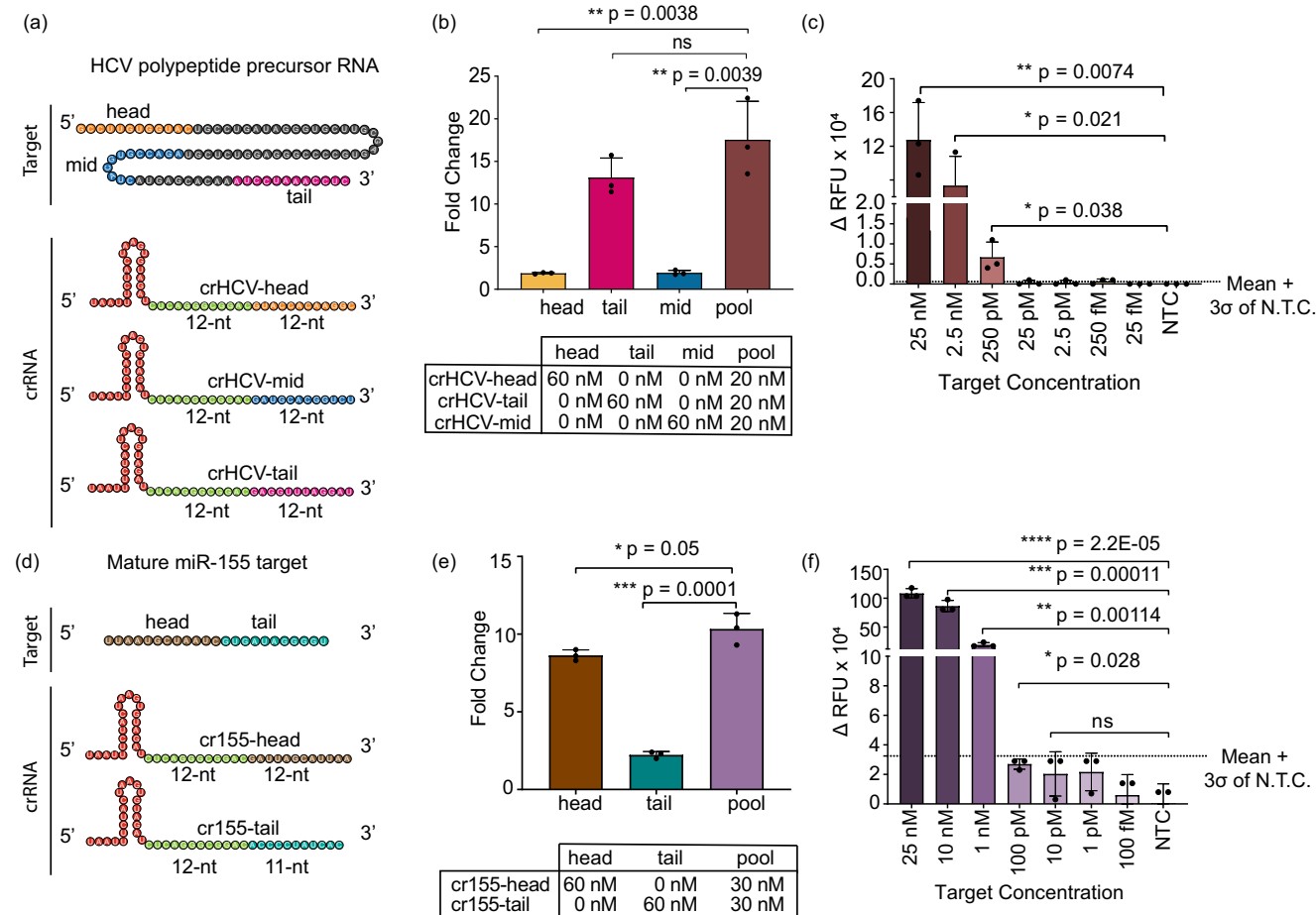

**Fig. 4 | Application of SAHARA for the detection of HCV and miR-155.**
**a** Schematic of an HCV polypeptide precursor RNA target and three crRNAs targeting it at different positions. Colors within the target and crRNA indicate the following: crRNA Pp region (green), head (orange), tail (purple), and middle (blue) sections of an HCV polypeptide precursor RNA. **b** Comparison among the head, tail, mid, and pooled HCV targeting crRNA. The plot represents the fold change in fluorescence intensity normalized to the NTC at $t = 60$ min ($n = 3$). **c** Limit of detection of HCV target using a pool of head, tail, and mid crRNA sequences. Plot represents the background subtracted fluorescence intensity at $t = 60$ min, for different concentrations of the target. **d** Head vs. Tail detection for a mature miRNA-155 target mediated by a split activator system. crRNAs were designed to target an S12 dsDNA GFP-activator in the Pp region and target either the head or tail region of an miR-155 target in the Pd region. **e** Comparison of normalized fluorescence intensity fold change values among cr155-Tail, cr155-Head, and a combination of both Head and Tail targeting crRNAs. **f** miR-155 limit of detection using a pooled crRNA with a split activator system. The plot represents the background subtracted fluorescence intensity at $t = 60$ min, for different concentrations of the target. All error bars represent the mean value +/− SD ($n = 3$). Statistical analysis for $n = 3$ biologically independent replicates comparing the fluorescence intensity at different target concentrations vs. the NTC was performed using a two-tailed $t$-test where ns = not significant with $p > 0.05$, and the asterisks (*$p \leq 0.05$, **$p \leq 0.01$, ***$p \leq 0.001$, and ****$p \leq 0.0001$) denote significant differences. ErCas12a was used for all experiments in this section. Source data are provided as a Source Data file.

infection that causes liver inflammation and can lead to serious complications such as liver damage, cirrhosis, and liver cancer. Early detection of HCV is important because it can help people receive treatment before the virus causes serious damage to the liver[39]. For HCV, we synthesized a short target RNA resembling a polypeptide precursor gene that is conserved across multiple HCV genotypes. For this RNA, we designed three different SAHARA guide RNAs each targeting it at either the 5′-end (head), 3′-end (tail), or in the middle (mid) (Fig. 4a).

The miRNA-155 (or miR-155) is known to play a crucial role in breast cancer progression and is overexpressed in breast cancer tissues[40,41]. Therefore, reliable detection of miR-155 is important for early diagnosis. The mature miR-155 is ~23-nt in length. We synthesized the mature miRNA target and designed two SAHARA guides targeting it at either 12-nt at the 5′-end or 11-nt at the 3′-end, keeping the S12-binding region constant (Fig. 4d).

For both miR-155 and HCV targets, we first tested detection with the different designed guides individually. While all the designed guides for both targets were functional and displayed *trans*-cleavage,

we observed significant variability in the activities produced by different guides. For miR-155, the head targeting guide showed an almost 5-fold higher activity than its tail targeting counterpart. Similarly, for HCV, the tail targeting guide showed as much as a 6-fold increase in activity as compared to head-targeting and the mid-targeting guide (Fig. 4b, e).

Upon closer inspection of the crRNAs with their corresponding targets, we observed that the secondary structure of the RNA target determines the activity of SAHARA (Fig. S13). Targets with a high amount of secondary structure are more inaccessible to bind to the Cas12a-crRNA complex, and are therefore harder to detect, while targets with relatively low or no secondary structure are detected easily. For miRNA-155, the fact that the tail guide only binds to 11-nt of the target as opposed to 12-nt binding in the head guide might play a role in the reduced activity.

It has previously been shown with the Cas13a enzyme that pooling together multiple crRNAs enhances the level of detection[42]. We rationalized that a similar approach would also work with SAHARA. In concordance with the previous reports[42], SAHARA also displayed a

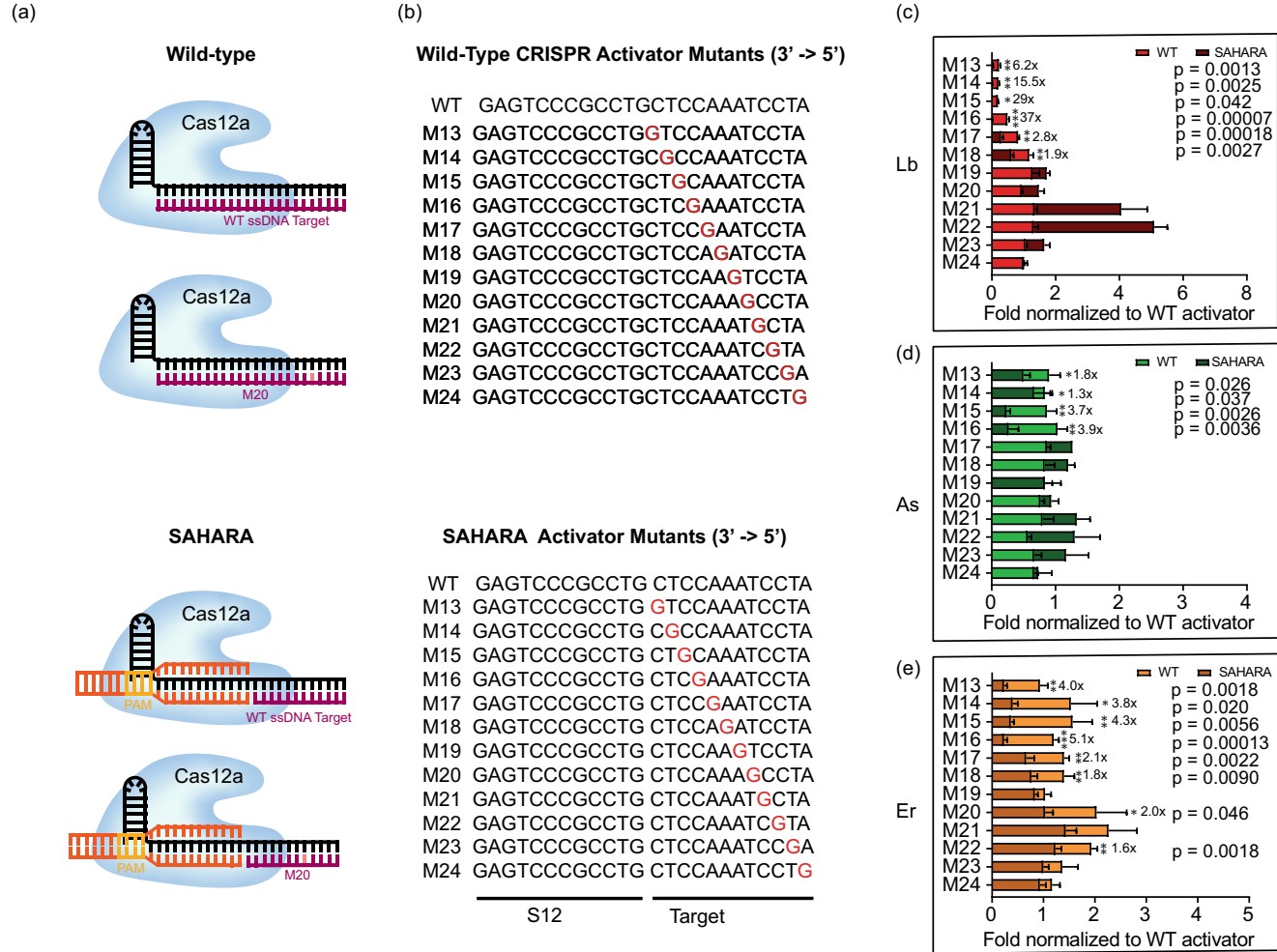

**Fig. 5 | Specificity of SAHARA towards single point mutations in target.**
**a** Schematic of WT vs. SAHARA CRISPR-Cas systems for the detection of a target nucleic acid. **b** ssDNA activators were designed with point mutations across the length of the activator. GFP-activator mutants were designed for a WT CRISPR activator (24-nt) and a SAHARA split activator system (12-nt + 12-nt). The mutation location is identified by 'M' following the nucleotide number where the base has been changed to guanine (3' to 5' direction). **c−e** Comparison of fold changes for the in vitro *trans*-cleavage assay between WT and SAHARA activator mutants

normalized to the WT activator for Cas12a orthologs (**c**: LbCas12a, **d**: AsCas12a, and **e**: ErCas12a). Comparison of RFU values at $t = 60$ min for the in vitro *trans*-cleavage assay between WT and SAHARA. Statistical analysis for $n = 3$ biologically independent replicates comparing the normalized fold change for the WT assay vs. SAHARA was performed using a two-tailed $t$-test where ns = not significant with $p > 0.05$, and the asterisks (*$p \leq 0.05$, **$p \leq 0.01$, ***$p \leq 0.001$, and ****$p \leq 0.0001$) denote significant differences. Error bars represent mean value +/− SD ($n = 3$). Source data are provided as a Source Data file.

higher activity with pooled crRNAs for both miRNA-155 and HCV targets (Fig. 4b, e).

Next, we tested to check the sensitivity of SAHARA. For this, we created dilutions of miRNA-155 and HCV targets ranging from 25 nM–10 fM and tested for the detection of each target at different concentrations with their corresponding pooled crRNAs. We obtained a limit of detection of 132 pM for HCV and 767 pM for the miRNA-155 RNA target (Fig. 4c, f). These results are in concurrence with earlier studies showing similar limits of amplification-free detection of ssDNA and dsDNA with Cas12a[43].

Finally, all the experiments performed for the HCV target in this section were done with an 86-nt long RNA. Since the length of the RNA seems to affect the performance of SAHARA, this RNA might not be a true representation of the length of targets that will be available for detection in a clinically relevant setting. To demonstrate detection with a more relevant RNA target, we ordered the Quantitative Synthetic RNA from Hepatitis C virus from ATCC (Cat# VR-3233SD) that consists of fragments from 5'UTR and X-tail region, thus being a more complex sample as compared to the previously used 86-nt RNA. Initially, we found the detection of this RNA target with our typical

protocol to be challenging due to the low concentration of this target RNA ($10^5$–$10^6$ copies/μL). However, we were able to detect this RNA upon increasing the concentration of crRNA, Cas12a, and S12 to 250 nM, 125 nM, and 105 nM respectively (Fig. S14).

## SAHARA improves mutation detection specificity

We hypothesized that SAHARA might be more sensitive in discriminating single point mutations in the target as compared to the WT CRISPR-Cas12a since SAHARA binds to significantly fewer nucleotides of the target. To test this, we designed 12-nt ssDNA activators with single-point mutations for detection with SAHARA, as well as 24-nt ssDNA activators with identical mutations for detection of WT CRISPR-Cas12a for comparison (Fig. 5a, b). We observed that the detection of single-point mutants with both SAHARA and WT CRISPR was position-dependent compared to the WT target. We used the mutant activators to compare the trans-cleavage activity of SAHARA with the activity of WT CRISPR-Cas12a for point mutations along the target. We observed that mutations in positions M13-M18 significantly decreased the SAHARA-mediated *trans*-cleavage activity for Lb and Er (Fig. 5c−e). The same was true with positions M13-M16 with As but not with positions

M14 and M15 (Fig. 5d). Mutations in positions beyond M18 did not decrease the *trans*-cleavage activity for either Cas12a ortholog as compared to the WT CRISPR counterpart. Our data indicated that SAHARA had improved specificity over WT-CRISPR for mutations near the split-activator boundary for the different Cas12a orthologs. This implied that mutations closer to the boundary of the split-activators have a larger effect on the *trans*-cleavage activity of SAHARA as compared to mutations away from the boundary.

To test the specificity of SAHARA in the context of background non-competing RNA, we took the 86-nt long HCV-RNA that we had previously tested in Fig. 4 and spiked it in a matrix consisting of either water or extracted nucleic acids from pooled healthy human serum samples. We compared the detection efficiency of the target RNA spiked in serum vs. target RNA in water using ErCas12a-based SAHARA.

Our results indicated that SAHARA was able to robustly detect the target RNA in both water as well as serum matrix. We observed that compared to water, there was a slight increase in the background activity for the RNA spiked in serum. We think this increase in background activity might be on account of non-specific interactions with similar RNA sequences in the serum matrix. However, we also observed that the on-target signal for the HCV RNA increased proportionally. Therefore, the fold-change of target detection in both water and serum remained the same, even though the overall fluorescence generated was higher in serum (Fig. S15).

In a diagnostic setting, depending on the kind of matrix being tested (blood, serum, saliva, urine, etc.) we recommend using a healthy matrix as a control to account for background activity arising from non-specific interactions within the matrix, and then making use of statistical methods to detect the presence or absence of target RNA based on the differences in the fluorescence intensity with respect to the control.

### S12 DNA can be repurposed as a switch to control *trans*-cleavage activity of SAHARA

To verify the PAM-dependency of SAHARA for initiating *trans*-cleavage, we designed three different S12 activators each containing either a TTTA, AAAT, or VVVN PAM respectively (Fig. 6a). TTTA is one of the canonical PAM for Lb and As Cas12a, while Er is canonically known to have a preference for YTTN PAM[44], AAAT is the PAM-complement (cPAM) sequence, and VVVN encompasses the space of all the PAM sequences that are not tolerated by Cas12a. We tested the detection of a short 20-nt RNA with these S12 activators. As expected, the TTTA-PAM S12 was able to mediate RNA detection with all three Cas12a orthologs (Fig. 6b–d). Upon changing the PAM from TTTA to AAAT or VVVN, the *trans*-cleavage activity was completely diminished for Lb and As. Interestingly, Er Cas12a was able to detect RNA even an AAAT PAM, hinting at a broader PAM tolerance for *trans*-cleavage than previously reported.

The GC content of the spacer region has been previously shown to affect the activity of Cas12a enzymes[45,46]. We conjectured whether the GC content of the S12 activator plays a role in the RNA detection activity of SAHARA. To check this, we designed different crRNAs and S12 activators consisting of GC content ranging from 25 to 75%. All 3 Cas12a orthologs were able to tolerate changes in the GC content for the activity of SAHARA (Fig. 6e–g). This implies that Cas12a orthologs can tolerate a wide range of GC content for RNA detection.

Finally, we tested to check the minimum concentration of S12 needed to initiate RNA detection with SAHARA. We varied the concentration of S12 from 0 to 1.5 nM in increasing amounts and tested for the detection of a 25 nM RNA target. We observed an increase in the *trans*-cleavage activity of different Cas12a orthologs with increasing S12 concentration thereby suggesting that the activity of SAHARA is S12-dependent (Figs. 6h–j, S16, S17). Surprisingly, S12 concentration as low as 50 pM concentration was sufficient to initiate *trans*-cleavage activity for a CRISPR-Cas12a complex consisting of 30 nM Cas12a and 60 nM crRNA. This suggests that a low amount of S12 is required to initiate the *trans*-cleavage activity of SAHARA. Remarkably, we observed that in the absence of S12, there was no *trans*-cleavage despite the presence of crRNA, Cas12a, and target. This suggested that S12 is critical for the initiation of *trans*-cleavage activity and can be used as a switch to selectively turn the activity ON or OFF.

### Multiplexed DNA and RNA detection with SAHARA

We could leverage the switch-like function of the seed-binding S12 DNA activator to selectively turn ON the *trans*-cleavage activity of an individual crRNA from a pool of multiple different crRNAs. To investigate this, we used three different crRNAs (crRNA-a, crRNA-b, and crRNA-c), with unique targets and S12 activators, and pooled them all together (Fig. 7a). To demonstrate that we can simultaneously detect DNA and RNA with SAHARA, we used an ssDNA target A for crRNA-a while targets B and C were ssRNAs. We then performed the detection of each of the three targets with the pooled crRNAs and different Cas12a orthologs, in a combinatorial fashion, in the presence or absence of the corresponding S12 (Fig. 7b–d).

In the no S12 control, there was no *trans*-cleavage activity with any of the target combinations despite the crRNA-Cas complex and the target being mixed, further reinforcing the idea that S12 is critical for activity with SAHARA. In the presence of different S12s, only the mixtures with the corresponding target and crRNA displayed *trans*-cleavage. For instance, in the presence of S12a, only the reactions where target A was available (A only, A + B, and A + B + C) were active, whereas the reactions without target A (B only, C only, or B + C) were inactive despite the Cas, crRNA-b, crRNA-c, and their respective targets being mixed. The data here shows strong evidence that the S12 DNA can be used to selectively activate specific crRNAs and can be used to control the *trans*-cleavage activity of CRISPR-Cas12a, thereby enabling simultaneous and multiplexable detection of both DNA and RNA targets.

Next, we postulated if we could perform multiplexed RNA detection by combining SAHARA with Cas13b[26]. We used DNA and RNA reporters consisting of orthogonal dyes to differentiate the signal obtained from SAHARA and Cas13. This is similar to the multiplexed detection with Cas13 and Cas12 demonstrated before[26], but here we aimed to detect RNA substrates with both Cas12 and Cas13, and not DNA. To test this, we used Lb, As, and Er Cas12a orthologs in conjunction with PsmCas13b to detect two distinct RNA targets (T1 and T2). We designed Cas12 and Cas13 guide RNAs such that the guide for Cas12 was complementary to only the T1 target while the guide for Cas13 was complementary to only T2 (Fig. 7e).

Upon performing the detection of the T1 and T2 targets individually and in conjunction, we observed that SAHARA produced *trans*-cleavage activity only in the presence of the T1 target, while Cas13b produced *trans*-cleavage only in the presence of T2 (Fig. 7f, g). Furthermore, the *trans*-cleavage signal obtained from SAHARA and Cas13b could be distinguished from each other by using orthogonal fluorescent dyes such as FAM and HEX on different types of reporter molecules. Thus, it is feasible to combine SAHARA with Cas13b for multiplexed detection of distinct RNA targets.

## Discussion

While CRISPR-Cas systems have been studied for over a decade[1], *trans*-cleaving Cas enzymes such as Cas12 and Cas13 are relatively new. Despite this, tremendous progress is being made towards studying and understanding their underlying mechanism, especially due to their utility in molecular diagnostics. Cas12a, in particular, has been extensively studied and widely used in DNA-based diagnostic platforms. However, its use has been limited to DNA detection, as it does not naturally tolerate RNA substrates. In this work, we have discovered that Cas12a's tolerance for RNA substrates is position-dependent on the crRNA. By binding RNA substrates at the PAM-distal end of the crRNA

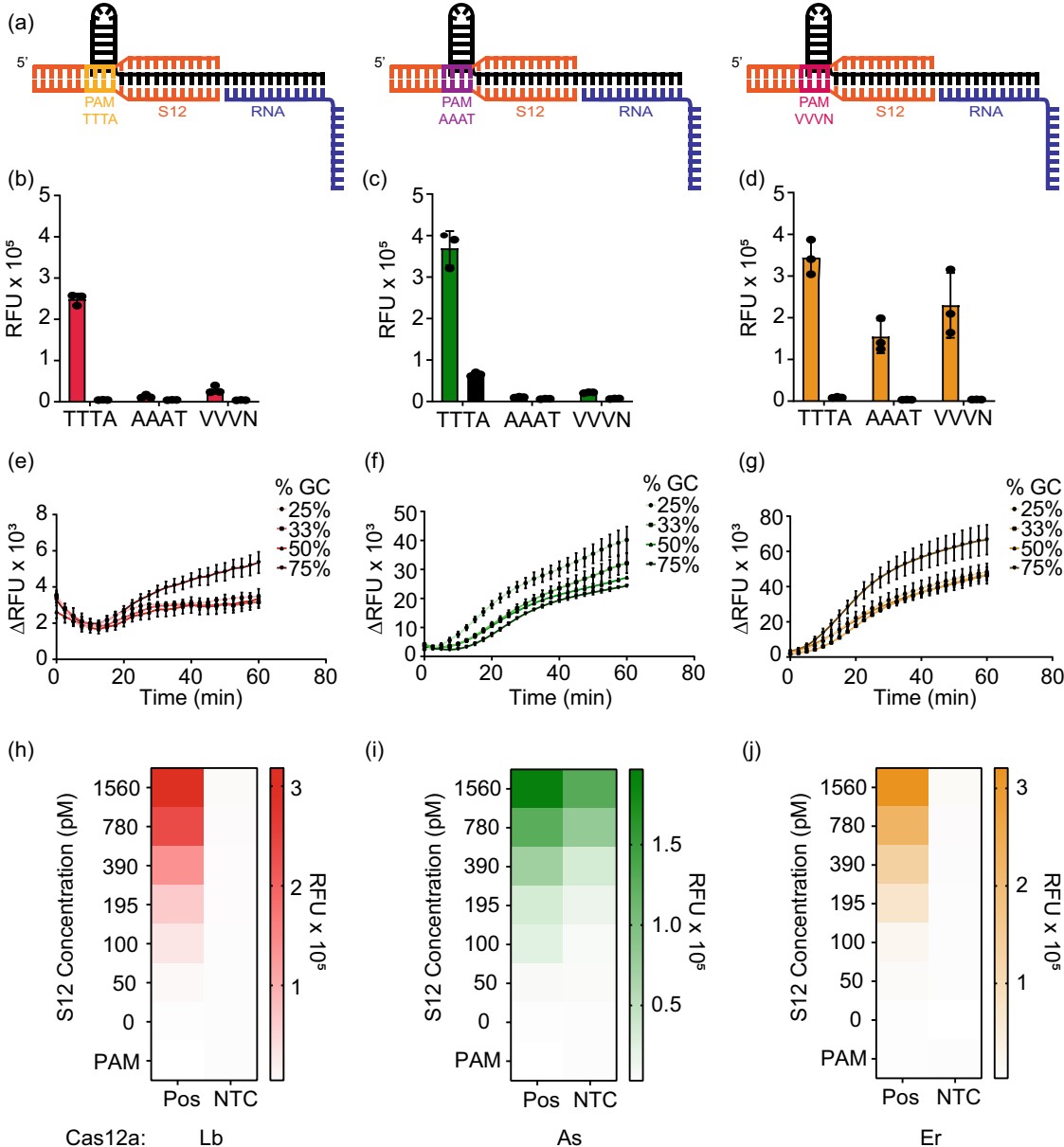

**Fig. 6 | Characterizing properties of the S12 DNA binding at the Pp region in SAHARA. a–d** PAM sequence tolerance of Cas12a orthologs (red = LbCas12a, green = AsCas12a, orange = ErCas12a) coupled with SAHARA. Comparison of *trans*-cleavage activity among S12 dsDNA activators containing different PAM sequences (*n* = 3). The PAM sequences TTTA, AAAT, and VVVN were assessed. The plot represents the fluorescence intensity in each group at time *t* = 60 min (*n* = 3). **e–g** Cas12a orthologs tolerate a wide range of GC contents in the crRNA and S12 dsDNA for RNA detection. The plots represent a graph of background subtracted fluorescence vs. time for the different groups (*n* = 3). **h–j.** The *trans*-cleavage activity of Cas12a with varying concentrations of S12 after incubation for 60 min at 37 °C. The heat map represents the fluorescence intensity at time *t* = 60 min. Error bars for all charts represent mean value +/− SD (*n* = 3). Source data are provided as a Source Data file.

and providing a DNA substrate at the PAM-proximal end, the trans-cleavage activity of Cas12a can be activated.

In contrast to Cas9, which uses two different active sites to generate a blunt double-stranded DNA break[47], Cas12a uses a single active site to make staggered cuts on the two strands of a dsDNA[6,12,16]. After cleaving the target DNA, Cas12a releases the PAM-distal cleavage product while retaining the PAM-proximal cleavage product bound to the crRNA[29,48]. This maintains Cas12a in a catalytically competent state, in which the active site of RuvC remains exposed to the solvent which then leads to *trans*-cleavage of neighboring single-stranded DNA molecules in a nonspecific manner.

It has been shown that crRNA-target DNA hybrids of length 14-nt or less do not trigger any *cis*- or *trans*-cleavage activity, and a crRNA-DNA hybrid of at least 17-nt is crucial for stable Cas12a binding and cleavage. These observations suggest that the interaction of Cas12a with crRNA-target DNA hybrid at positions 14–17 nt is critical for initiating cleavage[29,34,49]. Truncating the length of single-stranded DNA activators that bind to the crRNA significantly reduces the trans-cleavage activity of Cas12a. However, we found that using two truncated activators, each binding to a different region of the crRNA partially restores the lost activity.

Further experiments revealed that the PAM-proximal seed region of the crRNA exclusively tolerates DNA for initiating trans-cleavage, while the PAM-distal end can tolerate both RNA and DNA, except in the case of AsCas12a which interestingly tolerated RNA even at the Pp region. By designing long crRNAs with specific regions complementary to PAM-containing dsDNA and the target of interest, we developed a split-activator-based method called SAHARA for RNA detection with

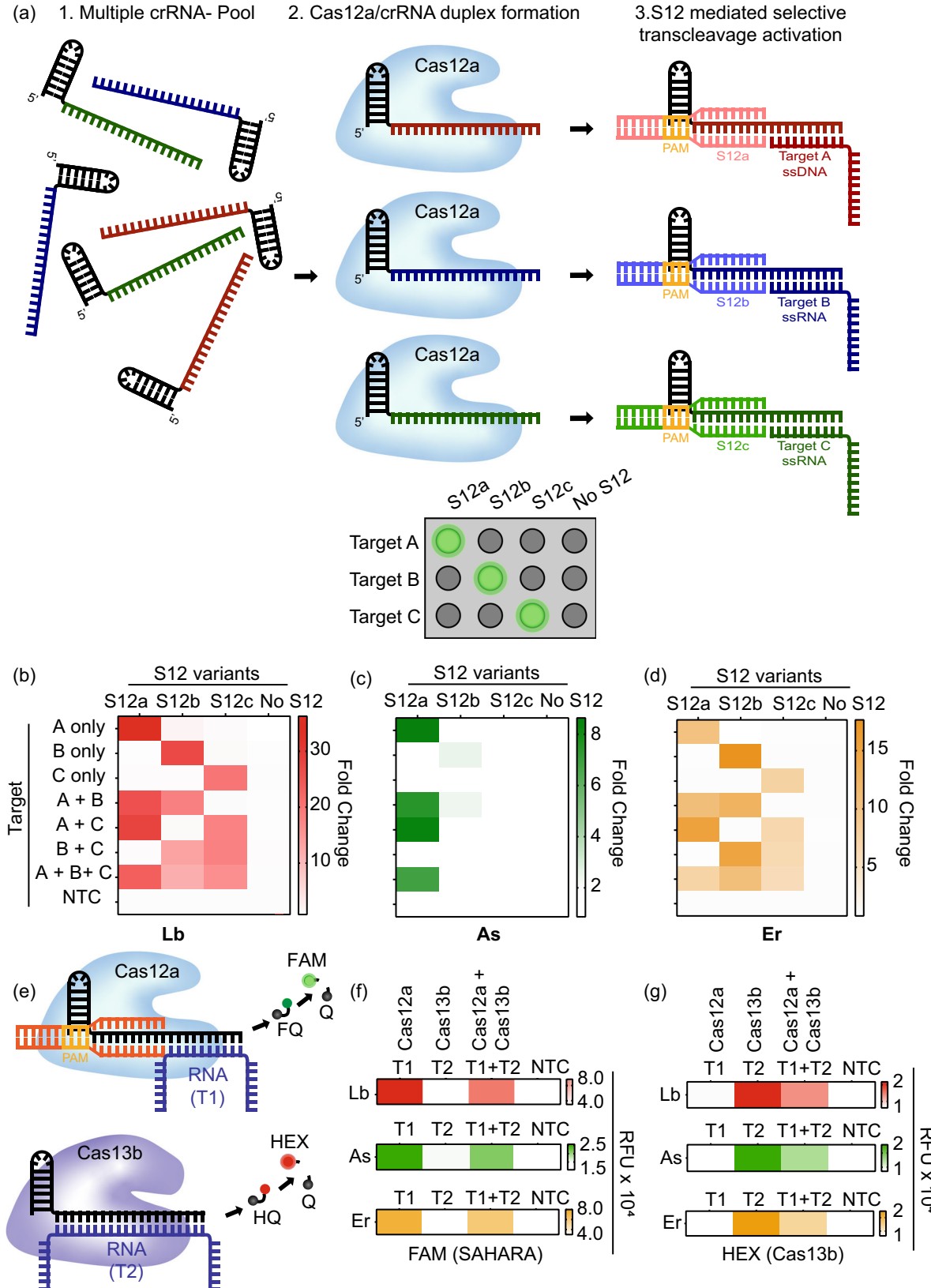

Cas12a. SAHARA enabled robust RNA detection activity for various lengths of RNA, with ErCas12a displaying the strongest activity for long RNA sequences.

We applied SAHARA for the detection of clinically relevant targets such as Hepatitis C virus (HCV) and miRNA-155. We found that the secondary structure of the target played an important role in the RNA detection activity of Cas12a. Increased secondary structure in the RNA target makes it more inaccessible to bind to the crRNA and reduces the activity. We also observed that pooling together multiple crRNAs targeting different regions of the activator

**Fig. 7 | Simultaneous detection of multiple targets with SAHARA. a** Schematic of multiplexed detection with SAHARA. A mixture of different crRNAs can be differentiated for *trans*-cleavage activity using sequence-specific S12 activators. **b–d** The heat maps depicting the *trans*-cleavage activity of 3 different pooled crRNAs (crRNA-a, crRNA-b, and crRNA-c) in the presence of 3 different S12 activators (S12a, S12b, S12c) or a no S12 control for Lb, As, and Er cas12a orthologs. Fold change compared to the No Target Control (NTC) at $t = 60$ min from the start of the reaction is plotted. 30 nM Cas12a, 60 nM crRNA, 25 nM S12 activators, and 25 nM of DNA or RNA targets were used in the assay ($n = 3$). **e** Schematic of multiplexed RNA detection with a combination of SAHARA and Cas13b. DNA or RNA reporters consisting of different colored dyes are used to distinguish the signal produced by Cas12a and Cas13b. **f, g** Multiplexed RNA detection using Lb, As, and Er orthologs of Cas12a in combination with PsmCas13b. Cas12a targets activator T1 and produces a signal in the FAM channel, while Cas13b targets activator T2 and produces a signal in the HEX channel. Heat map represents fluorescence intensity at time t = 60 min ($n = 3$). Source data are provided as a Source Data file.

enhanced the detection capability, as previously reported for Cas13. Applying this, we were able to detect picomolar levels of both miRNA-155 as well as HCV with SAHARA.

While SAHARA works well with synthetic RNA targets, a key limitation of our study is the lack of validation with respect to detecting RNA samples with more complex material in a clinical setting. A potential drawback of SAHARA is its specificity in more complex samples. Since SAHARA is designed to only bind to 12-nt of the target RNA, the odds of finding similar RNA sequences in a complex material are high, potentially resulting in false positives. Upon testing the detection of an HCV target spiked in nucleic acids extracted from healthy serum samples, we observed an increase in the background activity resulting from SAHARA as compared to the detection of the same sample in water. However, we also observed that the on-target activity increased proportionally.

In a diagnostic setting, depending on the kind of matrix being tested (e.g. blood, serum, saliva, urine, etc.) we recommend using a healthy matrix as a control to account for background activity arising from non-specific interactions within the matrix and then making use of statistical methods to detect the presence or absence of target RNA based on the differences in the fluorescence intensity with respect to the control.

Developing a robust, clinically applicable, diagnostic test with SAHARA will require additional efforts, especially with regards to improving not just the specificity but also the sensitivity of detection. SAHARA in its current form is limited to only picomolar levels of detection without amplification, which may not be good enough for the detection of clinical samples. However, a promising outlook lies in combining SAHARA with the plethora of amplification-free detection techniques[50,51] that can be game-changing for the field of Cas12a-based diagnostics.

## Methods

### Ethical statement
For this study, de-identified human serum collection and processing were approved as a non-human study by the University of Florida Institutional Review Board (IRB202003085), and all ethical regulations were followed.

### Serum samples
Serum samples from healthy patients were obtained from a commercial vendor, Boca Biolistics, procured from their network of CAP/CLIA accredited partner laboratories across the United States as remnant (leftover) samples. These samples were delinked under the IIRB delinking protocol SOP 10–00114 Rev E.

### Plasmid construction
Plasmids expressing Lb, As, and ErCas12a enzymes were constructed as described in Nguyen et al.[35]. Briefly, plasmids expressing LbCas12a and AsCas12a were obtained from Addgene (a gift from Zhang lab and Doudna lab) and directly used for protein expression. For ErCas12a, a plasmid containing the human codon-optimized Cas12a gene was obtained from Addgene, then was PCR amplified using Q5 Hot Start high fidelity DNA polymerase (New England Biolabs, Catalog #M0493S), and subcloned into a bacterial expression vector (Addgene plasmid #29656, a gift from Scott Gradia). The product plasmids were

then transformed into Rosetta™(DE3)pLysS Competent Cells (Millipore Sigma, Catalog #70956) following the manufacturer's protocols.

### Protein expression and purification
For protein production, bacterial colonies containing the protein-expressing plasmid were plated on an agar plate and grown at 37 °C overnight. Individual colonies were then picked and inoculated for 12 h in 10 mL of LB Broth (Fisher Scientific, Catalog #BP9723-500). The culture was subsequently scaled up to a 1.5 L TB broth mix and grown until the culture reached an OD = 0.6 to 0.8. The culture was then placed on ice before the addition of Isopropyl β- d-1- thiogalactopyranoside (IPTG) to a final concentration of 0.5 mM. The culture then continued to grow overnight at 16 °C for 14–18 hours.

The overnight culture was pelleted by centrifuging at 10,000 × g for 5 min. The cells were then resuspended in lysis buffer (500 mM NaCl, 50 mM Tris-HCl, pH = 7.5, 20 mM Imidazole, 0.5 mM TCEP, 1 mM PMSF, 0.25 mg/mL Lysozyme, and DNase I). The cell mixture was then subjected to sonication followed by centrifugation at 39,800 × g for 30 min. The cell lysate was filtered through a 0.22 μm syringe filter (Cytiva, Catalog #9913-2504) and then ran through into 5 mL Histrap FF (Cytiva, Catalog #17525501, Ni²⁺ was stripped off and recharged with Co²⁺) pre-equilibrated with Wash Buffer A (500 mM NaCl, 50 mM Tris-HCl, pH = 7.5, 20 mM imidazole, 0.5 mM TCEP) connected to BioLogic DuoFlow™ FPLC system (Bio-rad). The column was eluted with Elution Buffer B (500 mM NaCl, 50 mM Tris-HCl, pH = 7.5, 250 mM imidazole, 0.5 mM TCEP). The eluted fractions were pooled together and transferred to a 10–14 kDa MWCO dialysis bag. Homemade TEV protease (plasmid was obtained as a gift from David Waugh, Addgene #8827, and purified in-house)(44) was added to the bag, submerged in Dialysis Buffer (500 mM NaCl, 50 mM HEPES, pH 7, 5 mM MgCl₂, 2 mM DTT) and dialyzed at 4 °C overnight.

The protein mixture was taken out of the dialysis bag and concentrated down to around 10 mL using a 30 kDa MWCO Vivaspin® 20 concentrator. The concentrate was then equilibrated with 10 mL of Wash Buffer C (150 mM NaCl, 50 mM HEPES, pH = 7, 0.5 mM TCEP) before injecting into 1 mL Hitrap Heparin HP column pre-equilibrated with Wash Buffer C operated in the BioLogic DuoFlow™ FPLC system (Bio-rad). The protein was eluted from the column by running a gradient flow rate that exchanges Wash Buffer C and Elution Buffer D (2000 mM NaCl, 50 mM HEPES, pH = 7, 0.5 mM TCEP). Depending on how pure the protein samples were, additional size-exclusion chromatography may have been needed. In short, the eluted protein from the previous step was run through a HiLoad® 16/600 Superdex® (Cytiva, Catalog #28989335). Eluted fractions with the highest protein purity were selected, pooled together, concentrated using a 30 kDa MWCO Vivaspin® 20 concentrator, snap-frozen in liquid nitrogen, and stored at −80 °C until use.

### Target DNA, RNA, and guide preparation
All DNA and RNA oligos as well as the chimeric DNA/RNA hybrid crRNAs were obtained from Integrated DNA Technologies (IDT). Single-stranded oligos were diluted in 1xTE buffer (10 mM Tris, 0.1 mM EDTA, pH 7.5). Complementary oligos for synthesizing dsDNA were first diluted in nuclease-free duplex buffer (30 mM HEPES, pH 7.5; 100 mM potassium acetate) and mixed in a 1:5 molar ratio of

target:non-target strand. Both strands were then subjected to denaturation at 95 °C for 4 mins and gradient cooling at a rate of 0.1–25 °C.

For generating the 730-nt long GFP target sequence, Addgene plasmid pCMV-T7-EGFP (BPK1098) (Addgene plasmid # 133962, a gift from Benjamin Kleinstiver) was obtained and PCR amplified using Q5 Hot Start high fidelity DNA polymerase (New England Biolabs, Catalog #M0493S) from position 376-1125. The PCR amplified product was in-vitro transcribed using the HiScribe T7 High Yield RNA synthesis kit (NEB #E2040S) following the manufacturer's protocol. The transcribed product was treated with DNase I for 30 min at 37 °C and then purified using RNA Clean and Concentrator Kit (Zymo Research #R1016).

## Preparation of metal ion buffers
The different metal ion buffers were prepared by first creating a master mix of the following components: 50 mM NaCl, 10 mM Tris-HCl, and 100 µg/ml BSA. To this master mix, chloride salts of different monovalent, divalent, and trivalent cations ($NH_4^+$, $Rb^+$, $Mg^{2+}$, $Zn^{2+}$, $Co^{2+}$, $Cu^{2+}$, $Ni^{2+}$, $Ca^{2+}$, $Mn^{2+}$, and $Al^{3+}$) were diluted to a final concentration of 10 mM. The pH of the buffer was adjusted to 7.9 by adding 1 M NaOH.

## CRISPR-Cas12a reaction for fluorescence-based detection
All fluorescence-based detection assays were carried out in a low-volume, flat-bottom, black 384 well-plate. The crRNA-Cas12a conjugates were assembled by mixing them in NEB 2.1 buffer and nuclease-free water followed by incubation at room temperature for 10 min. The assembled crRNA-Cas12a mixes were then added to 250-500 nM FQ reporter and the necessary concentration of the target activator in a 40-µL reaction volume. The 384 well-plate was then incubated in a BioTek Synergy fluorescence plate reader at 37 °C for 1 hour. Fluorescence intensity measurements for a FAM reporter were measured at the excitation/emission wavelengths of 483/20 nm and 530/20 nm every 2.5 min. A final concentration of 30 nM Cas12a, 60 nM crRNA, and 25 nM of target activator are used in all the assays unless otherwise specified.

## Detection of HCV-RNA in serum
Healthy serum samples procured from Boca Biolistics were extracted using Quick-DNA/RNA Viral MagBead Kit (Zymo Cat# R2140). Synthetic RNA resembling a fraction of the HCV 5-UTR gene was spiked to a final concentration of 1 nM and tested with SAHARA using the fluorescence-based detection method.

## Detection of HCV-ATCC RNA
5-µL of HCV-ATCC RNA (Cat# VR-3233SD) was combined with 5-µL of CRISPR-Cas mix consisting of 250 nM crRNA, 125 nM Cas12a, 105 nM S12, and 500 nM of FQ reporter. The reaction was transferred to a 384-well plate and then incubated in a BioTek Synergy fluorescence plate reader at 37 °C for 1 h. Fluorescence intensity measurements for a FAM reporter were measured at the excitation/emission wavelengths of 483/20 nm and 530/20 nm every 2.5 min.

## Limit of detection calculation
To find the limit of detection (LoD), the *trans*-cleavage assay was carried out with several different dilutions of the activator. The LoD calculations were based on the following formula:[52]

$$LoD = \frac{3.3 \times Std\ of\ RFU\ in\ the\ absence\ of\ activator}{Slope\ of\ RFU\ vs.\ Activator\ concentration}$$

## Statistics and reproducibility
No statistical method was used to predetermine sample size. No data were excluded from the analyses. The experiments were not randomized. The Investigators were not blinded to allocation during experiments and outcome assessment.

## Reporting summary
Further information on research design is available in the Nature Portfolio Reporting Summary linked to this article.

## Data availability
All the data supporting the findings of this study are available within the Article and Supplementary Files. Source data is available in the Source Data file. The crystal structure of the LbCas12a enzyme (PDB ID: 5XUS) deposited by Yamano et al. can be obtained from Protein Data Bank (PDB) 5XUS. Source data are provided with this paper.

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

## Acknowledgements

We are grateful to the members of the Jain lab for their helpful discussions and the University of Florida (UF) Health Cancer Center for their support. Special thanks to Dr. Gary Wang and his lab members at the University of Florida for their insightful feedback on the project and Ms. Lilia G. Yang for her help with experiments and proofreading the manuscript. This work was financially supported by funds from the UF (P.K.J.), the UF Herbert Wertheim College of Engineering (P.K.J.), Shah Foundation Endowment Funds (P.K.J.), Florida Breast Cancer Foundation AGR00018466 (P.K.J.), NIH-NIAID R21AI156321 (P.K.J.), NIH-NIAID R21AI168795 (P.K.J.), and NIH-NIGMS R35GM147788 (P.K.J.). The funding sources did not have a role in the design of the study, the collection, analysis, or interpretation of data, nor in writing the manuscript.

## Author contributions

P.K.J., S.R.R., and S.S.A. conceived the idea and designed the experiments. S.R.R., E.K.V., G.M.S., S.S.A., L.S.S., K.S.M., and L.T.N. performed the experiments and data analysis. S.R.R., E.K.V., and G.M.S. wrote the initial manuscript. The manuscript was edited by P.K.J. and revised by all members.

## Competing interests

The authors declare the following competing interests: P.K.J., S.R.R., E.K.V., and S.S.A. are listed as inventors on a patent application related to the content of this work. Applicant: University of Florida. US Patent App. 63/726,074. Systems and methods for reverse transcription free RNA detection with CRISPR-Cas12a. Status: Patent pending. P.K.J. is a co-founder of Genable Biosciences, Par Biosciences, and CRISPR, LLC. The remaining authors declare no competing interests.
