## [Peer Review File · Nature Communications]

Reviewers' Comments:

Reviewer #1:

Remarks to the Author:

This manuscript reports previously undescribed specific RNA detection using CRISPR/Cas12a. The authors discovered this property by adding a short activating strand of crRNA-complimentary DNA near the 5' seed region. They subsequently described a very thorough characterization and investigation of this interesting behavior. The core discovery is largely unreported, and represents a significant advance in the understanding of the behavior, biology, and limitations of the Cas12a family of enzymes. However, a few minor issues with reporting and representation of the data should be addressed before acceptance of the manuscript, which are outlined below:

1. On page 6, in the last paragraph of the introduction as well as the results subheading on page 13, the authors claim SAHARA has improved detection specificity. The authors should be more specific in their claim of specificity since only point mutation tolerance was investigated. i.e. SAHARA has improved mutation detection specificity. To broadly claim improved specificity, the authors should provide data indicating SAHARA displays improved specificity over the WT crRNA in a complex sample containing numerous different RNAs. With a maximum Pd RNA length of 12 – 16 nts, specificity in complex samples would likely be reduced compared to longer WT crRNAs.
2. When describing the individual truncated targets in Fig 1, it is unclear where they align with the crRNA. More information is needed.
3. Fig. 2c shows that a short RNA sequence in the Pp region can also activate the trans-cleavage behavior of AsCas12a enzyme. What is the mechanism for that?
4. Fig. 3 shows AsCas12a providing significant short RNA detection with the WT crRNA that is comparable to or better than SAHARA with AsCas12a. Why is that? If it is true, then the theory of requiring a DNA substrate in the Pp region for RNA detection may not be true for all Cas12a enzymes.
5. Many graphs in Fig. 3 with n=6 data points appear to show distinctly bimodal distribution with respect to fluorescence intensity. What caused this distinct grouping of data that led to large standard deviation? Also, given the large standard deviation, more experimental repeats are needed for those conditions to confirm the assay signal level.
6. The authors claim that compared to using WT crRNA, SAHARA can be performed at room temperature. There are numerous reports of Cas12a target detection and trans-cleavage activity at room temperature. Additional data in Fig. S8 comparing use of WT crRNA vs. SAHARA at RT should be provided to highlight the improvement.
7. In the first paragraph of page 15, a reference to Fig. 6e-g should be added.
8. In the second paragraph of page 15, up to 1.5 nM S12 DNA was used in conjunction with 30 nM Cas12a and 60 nM crRNA. Why wasn't a more stoichiometrically equivalent amount of S12 DNA used?

Reviewer #2:

Remarks to the Author:

Rananaware et al. developed a novel CRISPR-based detection method, called SAHARA, which detects RNA (and DNA) substrates using the trans-cleavage activity of Cas12a (orthologues), without the need of reverse transcription or strand displacement. By providing a short ssDNA or a PAM-containing dsDNA to the seed region, SAHARA enables detection of RNA sequences at the PAM-distal end of the crRNA. The authors demonstrated that this method can be used for amplification-free detection of picomolar concentrations of miRNA155 and hepatitis C virus RNA. Moreover, the PAM-proximal DNA can be used as a switch to detect both DNA and RNA targets.

Lastly, the authors demonstrated that SAHARA is suitable for multiplexing and can be combined with the RNA-detecting Cas13 nuclease to generate two distinguishable signals.

Overall, this is an interesting manuscript describing a clever trick to detect RNA using the otherwise strictly DNA-targeting Cas12a nuclease. The strength of this manuscript is the multitude and thoroughness of the conditions tested and should be interesting for the readers with interest in CRISPR biology (for their findings on the apparent flexibility in target requirements for Cas12), but also for a wide, diagnostic-orientated audience. A major weakness of this study (as detailed below) is the lack of clarity in the presentation (and sometimes the description) of the results and the (lack of) discussing thereof.

Major comments

Although the authors convincingly demonstrated the use of their SAHARA technology on two clinically-relevant targets (miRNA155 and hepatitis C virus RNA), these experiments were conducted with synthetic RNA targets and not with real samples (e.g. isolated from patient material). Without requesting the authors to do this, it does pose the question how well SAHARA will perform in more complex sample material, especially with regards to the specificity. As SAHARA is limited to recognizing 'just' ~12 nucleotides of RNA, the odds of finding very similar RNA targets in more complex sample material seems rather high, potentially resulting false-positives. Results depicted in Figure 5 seem to suggest discrimination in single-nucleotide level, but it remains to be seen whether this holds up in more complex sample material. Furthermore, I have some reservations whether this is an inherent property of SAHARA or just of the wild-type Cas12 itself (see my comments below). The authors should at least discuss these potential limitations in the discussion.

The authors quantify the trans-cleavage activity as either fold-change, delta RFU or raw FRU in different parts of the manuscript. The authors should at least elaborate on the reasoning behind this, and why not being consistent with this throughout the manuscript, as it doesn't allow to make comparisons between the results.

Page 8 and Figures 2 and S3: The authors observed that PAM-containing dsDNA activators bolster trans-cleavage activity. To study the influence of the PAM, the authors conducted a number of experiments with different activator combinations, presented in Figure S3. However, the combinations tested are not adequate to address this. Firstly (judging from the schematics), the Pd substrates have the PAM at the wrong position (hence, it can never function as a PAM). Secondly, the authors compared a (Pp) PAM-containing substrate with a shorter substrate lacking nucleotides at the position of where the PAM would have been. A better comparison would be to test the same length substrate where the PAM is replaced by a non-PAM sequence.

Page 14: "We observed that mutations in positions M01-M06 significantly decreased the SAHARA-mediated trans-cleavage activity"

1. Since these mutations are not part of the seed sequence, this is quite unexpected. Is this consistent with other literature (especially since this effect also seems to be existent for the WT-Cas12)? Likewise, the authors also mention that "Mutations in positions beyond M06 did not decrease the trans-cleavage activity for either Cas12a ortholog", however, even the most seed-distal M012 mutation appears to impact trans-cleavage activity for both WT as SAHARA (Figure 5c).
2. The authors labeled this section "SAHARA improves the specificity of target detection", but the results seems to be quite similar to the wild-type Cas12 setup (and not consistent across the different orthologues tested). Perhaps a better conclusion would be to say that SAHARA has regained its sequence specificity (when compared to the wild-type Cas12 setup)?
3. The authors should consider switching their nucleotide numbering to a more conventional standard (where position 1 is the first base of the protospacer, base-pairing to the 5' most base of the spacer region of the crRNA, see for example Zetsche et al, doi.org/10.1016/j.cell.2015.09.038)

The discussion reads more like a repetition of the result section. In this reviewer's opinion, this

space could better be used for other matters, such as the benefits, limitations and outlook of their SAHARA technology, how it compares to the wealth of other Cas12-based diagnostic platforms that have been published before, etc.

Minor comments

Page 3: "or a PAM containing dsDNA"
Consider changing this to "PAM-containing"

Page 3: "Notably, SAHARA is Mg²⁺ concentration- and pH-dependent"
Not sure whether this is worth mentioning in the abstract

Page 4: "Cas (CRISPR-associated) proteins are RNA-guided endonucleases..."
This is not an accurate description of Cas proteins (i.e. not all Cas proteins are nucleases). Please adjust accordingly.

Page 4: "Cas12a is a class II,"
Please change this to "class 2"

Page 4: "...there are no reports of targeted RNA cleavage by Cas12a."
Please delete or rephrase, as this contradicts with the next sentence.

Page 5: "...also cleave RNA through the addition of a PAMmer sequence."
Given the relevance of this concept for the findings in this manuscript, it would be helpful (especially for a non-expert reader) if the authors could shortly explain what a PAMmer sequence is.

Results

Page 7: "...and is completely lost for activators less than 12-nt in length"
Looking at figures 1-d, perhaps the authors meant to say "less than 14-nt in length" here?

Page 7: "...and mimicking a full-length target, would be able to regain the lost trans-cleavage activity."
This is somewhat confusing, why "lost"? My guess is that the authors are referring to the lack of trans-cleavage activity when using single ≤ 14 -nt DNA activators; this should be explained better.

Page 9: "...that both ssDNA and ssRNA had significantly greater activity with the EN crRNA for LbCas12a"
Did the authors apply any statistical test to substantiate this claim?

Page 9: "This is consistent with previous observations with ENHANCE."
Please provide a reference.

Page 9 : "irrespective of the crRNA design being used. (Fig. 2f-h)."
Please delete the first full stop.

Page 10: "...somewhere between the seed region and the 3'-end of the crRNA protospacer"
Perhaps the authors meant spacer instead of protospacer here? Consider rephrasing this to "3' spacer-end of the crRNA". Please adjust "crRNA protospacer" on page 11 accordingly.

Consider moving the paragraph "While a fully complementary RNA activator...can detect both RNA and DNA." (testing the limits of RNA detection) after the text describing the results of Figure 2 (showing that RNA can be detected).

Page 11: The sentence "...and discovered that most metal ions including severely inhibited the trans-cleavage activity of SAHARA" seems to be incomplete (including what?).

Page 11: "...consistent with the literature."

Please cite the literature that is being referred to here.

Page 12: "targeting it at either 12-nt at the 5'-end or 11-nt at the 3'-end, keeping the S12-binding region constant"

The authors should refer to Figure 4d here.

Page 12: "SAHARA also displayed a higher activity with pooled crRNAs for both miRNA-155 and HCV targets."

The authors should refer to the corresponding figures here.

Page 14: "TTTA is one of the canonical PAM for Cas12a"

The authors might want to be a bit more clear on what the tolerated PAMs for the different Cas12a orthologues are. For instance, ErCas12 has been reported to recognize YTTN PAMs (10.1089/crispr.2019.0026), potentially explaining why "Er Cas12a was able to detect RNA even with an VVVN PAM".

In this reviewer's opinion, the "S12 DNA can be repurposed as a switch to control the trans-cleavage activity of SAHARA" is a bit unnecessary, as many of the outcomes are to be expected, namely that a) non-canonical PAMs fail to stimulate trans-cleavage activity, b) that GC-content has no influence (why would it?) and c) that S12 is critical for the initiation of trans-cleavage activity (which can already be concluded from the results in Figure 3).

Page 15: "we used an ssDNA target A for crRNA-a while targets B and C were ssRNAs"

Please be consistent when referring to the targets (in the text they are called targets A,B,C, whereas in Figure 7 they are called targets 1,2,3). Also indicate in the figure whether a target is RNA or DNA.

Page 15: "...in the presence or absence of the corresponding S12. We also used a control wherein no S12 sequence was supplied to the crRNA-Cas-target mix."

This is ambiguous, is there a difference between "absence of ...S12" and "no S12 sequence"??

There are no references to Figures 7a-e throughout the whole manuscript!

Page 16: "We also used a FAM-containing DNA reporter and a HEX-containing RNA reporter to distinguish the signals of SAHARA and Cas13b." seems to be repeated just a few lines down:

"Furthermore, the

trans-cleavage signal obtained from SAHARA and Cas13b could be distinguished from each other by using orthogonal fluorescent dyes such as FAM and HEX on different types of reporter molecules"

Page 17: "While CRISPR-Cas systems have been studied for almost a decade, trans-cleaving Cas enzymes such as Cas12 and Cas13 are relatively new"

CRISPR-Cas systems have been studied well over a decade, starting in 2007:

(doi.org/10.1126/science.1138140) and trans-cleaving activity has already been shown in 2018 (doi.org/10.1126/science.aar6245)

Page 19: "all three cas12a"

Change to "all three Cas12a"

The discussion could benefit from making reference to the figures that are being discussed.

Figures and Legends

Figure 1b-d: it is unclear how the Pp and Pd DNA activators are truncated and designed, are these all 3' truncations (or 3' truncations for Pd and 5' for Pp?), what regions are part of the protospacer and/or seed sequence, where is the transition between Pp and Pd? etc. In addition, the authors should explain what is meant by "NTC". Without this information, the data is very hard to interpret. The authors should clarify this the figure and text.

Figure 1e-g. It is unclear why target combinations with a total length longer than 20 nt (e.g. 14-nt Pp + 8-nt Pd) seem to perform worse compared to their 20-nt counterparts. Do the authors have an explanation for this? Similar to my previous comment, it is also unclear how the >20-nt version are extended (at the 5' for the Pd ones?).

Figure legend 1b-d: "Fold change at t=60 minutes of in vitro trans-cleavage assay..."
Please revise (what does fold change of an assay mean, fold change in what, compared to what?)

Figure 3a: please indicate in the legend that the red part of the long RNA represents the target sequence.

Figures 3e-g: This is quite confusing. 1) What RNA was used as a target and why not both as in panels b-d? 2) Is "SR-12" the same condition as "SAHARA" (in panels b-d)? If so, why not label them the same? 3) If both the "SR-12" and "WT" conditions are the same as in panels b-d, why not add the scrambled data there? It seems a bit excessive to spend 3 panels on data showing that the scrambled guide RNA doesn't result in trans-cleavage activity.

Figure S6: What RNA target was used? More importantly, since the authors wanted to test whether longer (>20 nt) crRNA perform better, why was the canonical 20 nt crRNA not included as a benchmark?

Figure S7: what does "w.r.t." stand for?

Figure 5: It is a bit confusing that the orientation of the Cas12 schematics has been mirrored when compared to the previous schematics, which concomitantly forced the authors to depict sequences in the 3' to 5' orientation. Why not keep it consistent? Also, please indicate which Cas12a orthologue was used in the figure.

Reviewer #3:

Remarks to the Author:

Rananaware et. al. describe the ability of three orthologs of CRISPR-Cas12a to detect single-stranded DNA and RNA targets when a small 12 nt duplex containing a PAM and seed is also present. The method described by the authors (SAHARA) expands the utility of Cas12a to detect RNA without the need for reverse transcription steps. The authors also demonstrate the potential of SAHARA in multiplexed detection of both DNA and RNA targets in pooled samples.

This work reveals an interesting functionality of Cas12a that was not previously realized that could be used to expand Cas12a-based diagnostics. However, more needs to be done to establish SAHARA as a viable RNA diagnostic tool.

My primary concern is that the authors did not validate RNA diagnostic activity in the context of what would be reflected in a true disease diagnosis situation. For example, the HCV RNA used in this study was only 86 nt long. The validation experiment should be done with the entire HCV genome and with background RNA that would normally be present in a patient sample. Indeed, AsCas12a was not able to distinguish long RNAs from background in a previous experiment, suggesting the length of the RNA can influence the ability to detect RNA. Additionally, the authors did not indicate which ortholog of Cas12a was used to do the validation tests.

The work performed to evaluate specificity compared point mutants of a single stranded DNA target to a completely complementary target. Although the data revealed some differences in mismatch tolerance between the SAHARA and WT the authors should evaluate how mutations in RNA targets influence the diagnostic activities as ssDNA substrates will not be available in patient diagnostic samples. The authors should also evaluate specificity in the presence of competing non-target RNA and the RNA targets used should be of a length similar to what would be observed in a natural sample instead of the exact complement to the split guide.

Other concerns and suggestions:

The authors should consider presenting the Cas12a with the crRNA hairpin and PAM-binding region on the left side of their Cas12a diagrams as shown in figure 5. The inverse of this depiction with the 3' end of the guide RNA on the left, shown in figures 1.,2.,3., 6., and 7. is opposite of how Cas12a is usually depicted schematically in the field. This inverse orientation becomes particularly problematic in figure 3, when the PAM sequence is indicated in panel A. The canonical 5'-TTTA-3' PAM indicates the PAM sequence reading left to right, but the way the PAM is depicted it should be inverted to 3'-ATTT-5' to match with the orientation of the PAM in the diagram.

Additionally, crystal structures indicate the PAM remains double stranded when bound by Cas12a. However, several figures depict Cas12a bound to a duplex with an unwound PAM. A more accurate diagram of Cas12a bound to a PAM containing duplex should be diagrammed instead.

To better reflect the design of the dsDNA duplex, in the abstract consider replacing the phrase "a PAM containing dsDNA to the seed region" with "a dsDNA spanning PAM and seed regions."

The statement "Cas (CRISPR-associated) proteins are RNA-guided endonucleases that when complexed with the CRISPR RNAs" is somewhat inaccurate as (i.) not all Cas proteins are endonucleases, and (ii) not all Cas proteins complex with crRNA.

Usually, Class 2 CRISPR systems are indicated with a Hindu-Arabic numeral 2 instead of roman numeral II.

Can the authors address whether split targets (both DNA and RNA targets) are still cleaved in cis when they are long enough to fit into the RuvC endonuclease site?

The placement of significance brackets in figure 1.b-d makes it unclear which data are being compared. It seems as if comparisons are between Pd data sets. Is this correct? It should be made more clear in the figure legend what data sets are being compared.

Can the authors address what is meant by "fold change" in figures 1.b-g, 2.b-d, 7.b-d. It is unclear what the baseline is for fold comparison.

Could the authors include statistical comparisons with data presented in figure 2.f-h? Additionally, plotting these data on a Log scale could help significant differences be more apparent.

Can the authors explain why a dashed line is included in figure 3.b-d at 10×10^5 ?

The sentence "To test these targets, we designed two crRNAs – a canonical WT as well as a SAHARA crRNA that only binds to the target at the Pd end while binding to a synthetic S12 at the Pp-end." Suggests SAHARA crRNAs are different from WT crRNAs. Could the authors provide more details on how the sequences differ in the region that would bind the S12 duplex?

In figure 4, the limit of detection is indicated by a dashed line in figure 4.c. It would be appropriate to include a similar line indicating limit of detection to figure 4.f

In some figures, only the colors are used to symbolize the different Cas12a orthologs (Lb, As, and Er as Red, Green and Orange, respectively). The authors should indicate with text in each figure and in each figure legend which orthologs are used to generate the data in case the figures are printed in black and white and to accommodate color-blind individuals.

Change "Pp 10-nt nucleotides of the crRNA protospacers" to "Pp 10-nt nucleotides of the crRNA spacer"

The sentence "To study the effect of different metal ions on the activity of SAHARA, we tested a range of different metal cations (NH_4^+ , Rb^+ , Mg^{2+} , Zn^{2+} , Cu^{2+} , Co^{2+} , Ca^{2+} , Ni^{2+} , Mn^{2+} , and Al^{3+}) and discovered that most metal ions including severely inhibited the trans-cleavage activity

of SAHARA (Fig.S9).” The word “metal” should be removed as NH_4^+ is not a metal ion, and the word “including” should be removed to clarify the meaning of the sentence.

Reviewer #4:
None

AUTHORS' RESPONSE TO THE REVIEWERS:

*We sincerely thank all the reviewers for their time and efforts in thoroughly reading the manuscript and providing detailed comments as well as their expert insights. We believe our manuscript is now in a lot better shape due to their inputs, especially in terms of representing the data. We have addressed all the comments below with *responses marked in blue* and changes to the manuscript highlighted in *yellow*.*

REVIEWER #1 (REMARKS TO THE AUTHOR): R1

This manuscript reports previously undescribed specific RNA detection using CRISPR/Cas12a. The authors discovered this property by adding a short activating strand of crRNA-complimentary DNA near the 5' seed region. They subsequently described a very thorough characterization and investigation of this interesting behavior. The core discovery is largely unreported and represents a significant advance in the understanding of the behavior, biology, and limitations of the Cas12a family of enzymes. However, a few minor issues with reporting and representation of the data should be addressed before acceptance of the manuscript, which are outlined below:

R1.1 On page 6, in the last paragraph of the introduction as well as the results subheading on page 13, the authors claim SAHARA has improved detection specificity. The authors should be more specific in their claim of specificity since only point mutation tolerance was investigated. i.e. SAHARA has improved mutation detection specificity. To broadly claim improved specificity, the authors should provide data indicating SAHARA displays improved specificity over the WT crRNA in a complex sample containing numerous different RNAs. With a maximum Pd RNA length of 12 – 16 nts, specificity in complex samples would likely be reduced compared to longer WT crRNAs.

Response: We appreciate the reviewer's feedback. We agree with the reviewer's comment and have changed our claim to be made specific to point mutation tolerance (i.e., SAHARA displays improved specificity towards detection of point mutations) wherever relevant.

In order to test the specificity of SAHARA in complex samples we spiked the 86-nt long HCV-RNA that we had previously tested in Fig. 4 to a matrix consisting of extracted nucleic acids from healthy human serum samples. We compared the detection efficiency of the target RNA spiked in serum vs. target RNA in water. We used ErCas12a for this experiment.

Our results indicated that SAHARA was able to robustly detect the target RNA in both water as well as serum matrix (Fig. S15). We observed that compared to water, there was a slight increase in the background activity for the RNA spiked in serum. However, we also observed that the on-target activity increased proportionally. Therefore, the fold-change of target detection in both water and serum remained the same.

Fig S15: Target HCV RNA was spiked in a matrix consisting of either water or nucleic acid extract from healthy serum sample to a final concentration of 1 nM. The detection efficiency of target RNA spiked in water or serum was determined with ErCa s12a based SAHARA. (a) Plot representing RFU of target RNA detection in water or serum at t=60 min. Error bars represent S.D. (n=4) (b) Plot representing fold change of fluorescence intensity from target RNA sample with respect to the fluorescence from the no target control (NTC) in water and serum. Error bars represent S.D. (n=4). Statistical analysis was performed using a two-tailed t-test where ns = not significant with $p > 0.05$, and the asterisks (* $P \leq 0.05$, ** $P \leq 0.01$, *** $P \leq 0.001$, **** $P \leq 0.0001$) denote significant differences.

R1.2 When describing the individual truncated targets in Fig 1, it is unclear where they align with the crRNA. More information is needed.

Response: We thank the reviewer for the feedback. We have modified the schematic for Fig. 1 to include more information regarding how the individual activators align with the Pp and Pd region of the crRNA. We have also included a supplementary figure which depicts how the truncated activators of the different length bind to Pp and Pd.

Fig. 1: Cas12a orthologs tolerate short ssDNA activators (6-12 nt) when added in combination. Schematic representation of a crRNA-Cas12a complex performing *trans*-cleavage of ssDNA reporters following the recognition of two split-activators. **b-d.** Fold change with normalized to No Target Control (NTC) at t=60 minutes of *in vitro trans*-cleavage assay with Cas12a orthologs (red = LbCas12a, green = AsCas12a, orange = ErCas12a) activated by individual truncated ssDNA activators of length 6-20 nt. Statistical comparisons are made for a combination of both Pp and Pd data sets between the different lengths of the targets. Statistical analysis was performed using a two-tailed t-test where ns= not significant with $p > 0.05$, and the asterisks (* $p \leq 0.05$, ** $p \leq 0.01$, *** $p \leq 0.001$, and **** $p \leq 0.0001$) denote significant differences. **e-g.** Heat maps representing fold change with respect to NTC at t=60 minutes of an *in vitro trans*-cleavage assay activated by combinations of truncated ssDNA activators of different lengths ranging from 6-14 nt in the Pp and Pd regions. The reactions contained 25 nM truncated ssDNA GFP-activators, 60 nM Cas12a, and 120 nM crGFP and were incubated for 60 min at 37°C. The NTC represents the condition when neither the Pp nor the Pd activator is present in the reaction. Error bars represent SD (n=3).

Fig. S3: Schematic depicting how truncated ssDNA activators of different length bind to the PAM-proximal (Pp) and PAM-distal (Pd) regions of the crRNA.

R1.3 Fig. 2c shows that a short RNA sequence in the Pp region can also activate the trans-cleavage behavior of AsCas12a enzyme. What is the mechanism for that?

Response: We thank the reviewer for the question. We were quite surprised to observe this unusual behavior of AsCas12a. We are not quite sure of the mechanism at this stage, however, so far in our investigation only AsCas12a seems to be capable of tolerating RNA substrates at the Pp region to activate trans-cleavage. Other Cas12a orthologs, including Lb and Er, show a strict preference for DNA only. In addition, we tested 3 other orthologs to investigate this mechanism, please refer to R1.4 below for additional data.

Interestingly, in our experiments with chimeric DNA-RNA hybrid crRNAs, only AsCas12a was able to completely tolerate the chimeric crRNA without any loss in activity (Fig. S5). Taken together, these experiments suggest that AsCas12a might have some fundamental mechanistic differences in its ability to initiate trans-cleavage as compared to other Cas12a orthologs.

Fig S5: Chimeric DNA-RNA guides complexed with Cas12a. (a) Schematic representation of chimeric DNA-RNA hybrid crRNAs complexed with Cas12a and activated with WT ssDNA activators. Chimeric crRNA was designed by changing 12-nt near the PAM-proximal 5'-end of the crRNA to DNA (12D8R crRNA) and changing the PAM distal 8-nt end of the crRNA to DNA (12R8D crRNA). WT crRNA is represented on graphs b-d by triangles, 12R8D crRNA is represented by squares, and 12D8R crRNA is represented by circles. (b-d) Relative RFU values of in vitro *trans*-cleavage assay with Cas12a orthologs (Lb- red, As- green, Er- orange) complexed with WT crRNA, 12D8R crRNA, and 12R8D crRNAs. (e-g) Fold change at 60 min is represented for each crRNA and three Cas proteins. The reactions contained 25 nM ssDNA GFP WT activator, 60 nM Cas12a, and 12 nM crRNA (WT, 12R8D, 12D8R). Reactions were incubated for 60 min at 37°C. Error bars represent SD (n=3).

R1.4 Fig. 3 shows AsCas12a providing significant short RNA detection with the WT crRNA that is comparable to or better than SAHARA with AsCas12a. Why is that? If it is true, then the theory of requiring a DNA substrate in the Pp region for RNA detection may not be true for all Cas12a enzymes.

Response: We thank the reviewer for the question. As explained in the previous comment, in our observations with multiple Cas12a orthologs so far, AsCas12a seems to be the only one capable of tolerating RNA substrates at the Pp region. To investigate the tolerance of Pp RNA detection with other different Cas12a, we tested three new orthologs – Hk, Mb, and Bs - that were not previously described in this study. We had previously determined that Hk is phylogenetically closely related to As¹, so we were curious to see if it would show similar tolerance towards Pp RNA. Furthermore we included two other understudied orthologs, Mb and Bs, for comparison as they were readily available in the lab from our previous work in which we had characterized their trans-cleavage activity¹.

Fig. S4: Heat map representing the fold change in fluorescence with respect to the NTC for a split-activator combination of 10-nt PAM-distal (Pd) DNA and either a 10-nt PAM-proximal (Pp) RNA or an equivalent Pp DNA. The experiments were performed with 6 different orthologs of Cas12a – Lb, As, Er (left) and Hk, Mb, Bs (right).

Upon testing the new orthologs with a 10-nt Pp RNA or equivalent Pp DNA (with a 10-nt Pd DNA in each case), we observed that all three orthologs displayed trans-cleavage activity only with Pp DNA but not Pp RNA. Suggesting, that unlike AsCas12a, these Cas12a also have a strict preference for DNA only at the PAM-proximal region.

This unusual characteristic of AsCas12a might be playing a role in its ability to directly detect 20-nt RNA substrates with a WT crRNA. Currently, we don't know the mechanism for this behavior. However, if you refer to Fig. 3c, the ability of AsCas12a to detect RNA substrates directly with a WT crRNA greatly diminishes for longer lengths of RNA. AsCas12a is known to have one of the highest trans-cleavage activities amongst known Cas12a orthologs. Maybe this behavior is an artifact of AsCas12a's innately high ability to do trans-cleavage as well as the fact that short RNAs are easier to bind to, on account of a low degree of secondary structure. This is just speculation on our part because LbCas12a, which also demonstrate high trans-cleavage activity, lacks this characteristic behavior. If other Cas12a orthologs with a similar RNA tolerance at the Pp region are discovered, it would be quite interesting to compare them to AsCas12a and study what makes them different from the Pp DNA-only Cas12a orthologs.

R1.5 Many graphs in Fig. 3 with n=6 data points appear to show distinct bimodal distribution with respect to fluorescence intensity. What caused this distinct grouping of data that led to large standard deviation? Also, given the large standard deviation, more experimental repeats are needed for those conditions to confirm the assay signal level.

Response: The bimodal distribution arises from the fact that for the data in this figure, two separate biological replicates were performed at different times, each with three technical replicates, and raw fluorescence intensities are reported. To make data visualization clearer, we have color coded the data points from the two different experiments separately.

Fig. 3: Development of SAHARA for the detection of wide range of RNA targets a. Schematic representation of Cas12a complexed with WT vs. SAHARA crRNA and activated by either a short (20-nt) or long (730-nt) RNA activators. The orange section of the long RNA activator corresponds to the target sequence. b-d. Comparison of *trans*-cleavage activity among Cas12a orthologs for the short vs. long combinatorial schemes seen in (a). The plot represents raw fluorescence units (RFU) plotted for time t=60 min. For this experiment, two biological replicates each with three technical replicates were performed. The data points for the two biological replicates are represented by black and gray, respectively. e-g. Detection of the long RNA target with SAHARA by using either a non-targeting scrambled S12 (SR-Scr) or a targeting S12 (SR-S12). Plot represents RFU at t=60 min. All error bars represent SD (n=3). Statistical analysis was performed using a two-tailed t-test where ns = not significant with $p > 0.05$, and the asterisks (* $P \leq 0.05$, ** $P \leq 0.01$, *** $P \leq 0.001$, **** $P \leq 0.0001$) denote significant differences.

R1.6 The authors claim that compared to using WT crRNA, SAHARA can be performed at room temperature. There are numerous reports of Cas12a target detection and trans-cleavage activity at room temperature. Additional data in Fig. S8 comparing use of WT crRNA vs. SAHARA at RT should be provided to highlight the improvement.

Response: We appreciate the reviewer for the feedback. We would like to clarify that we are not claiming improved detection at room temperature with SAHARA compared to WT. We merely want to demonstrate that SAHARA can also work at room temperature, if necessary, for the application.

R1.7 In the first paragraph of page 15, a reference to Fig. 6e-g should be added.

Response: We thank the reviewer for pointing that out. We have added a reference to Fig. 6e-g on page 15.

R1.8. In the second paragraph of page 15, up to 1.5 nM S12 DNA was used in conjunction with 30 nM Cas12a and 60 nM crRNA. Why wasn't a more stoichiometrically equivalent amount of S12 DNA used?

Response: We thank the reviewer for the question. We tested S12 DNA concentrations all the way up to 50 nM (see Supplementary Fig. S13, reproduced below for ErCas12a).

Fig. S16: Effect of S12 concentration ranging from 50 nM -780pM on the trans-cleavage activity of SAHARA is shown for Cas12a. The plot of RFU at t=60 min in the presence or absence of 25 nM target HCV-RNA and different S12 concentrations is shown. Error bars represent S.D. (n=3).

However, we noticed that at concentrations of S12 above 1.5 nM, non-specific trans-cleavage activity even without the target RNA would get activated. It seems, stoichiometrically, only a small concentration of the S12 DNA is necessary to perform SAHARA.

REVIEWER #2 (REMARKS TO THE AUTHOR): R2

Rananaware et al. developed a novel CRISPR-based detection method, called SAHARA, which detects RNA (and DNA) substrates using the trans-cleavage activity of Cas12a (orthologues), without the need of reverse transcription or strand displacement. By providing a short ssDNA or a PAM-containing dsDNA to the seed region, SAHARA enables detection of RNA sequences at the PAM-distal end of the crRNA. The authors demonstrated that this method can be used for amplification-free detection of picomolar concentrations of miRNA155 and hepatitis C virus RNA. Moreover, the PAM-proximal DNA can be used as a switch to detect both DNA and RNA targets. Lastly, the authors demonstrated that SAHARA is suitable for multiplexing and can be combined with the RNA-detecting Cas13 nuclease to generate two distinguishable signals.

Overall, this is an interesting manuscript describing a clever trick to detect RNA using the otherwise strictly DNA-targeting Cas12a nuclease. The strength of this manuscript is the multitude and thoroughness of the conditions tested and should be interesting for the readers with interest in CRISPR biology (for their findings on the apparent flexibility in target requirements for Cas12), but also for a wide, diagnostic-orientated audience. A major weakness of this study (as detailed below) is the lack of clarity in the presentation (and sometimes the description) of the results and the (lack of) discussing thereof.

R2.1 Although the authors convincingly demonstrated the use of their SAHARA technology on two clinically relevant targets (miRNA155 and hepatitis C virus RNA), these experiments were conducted with synthetic RNA targets and not with real samples (e.g. isolated from patient material). Without requesting the authors to do this, it does pose the question how well SAHARA will perform in more complex sample material, especially with regards to the specificity. As SAHARA is limited to recognizing 'just' ~12 nucleotides of RNA, the odds of finding very similar RNA targets in more complex sample material seems rather high, potentially resulting false positives. Results depicted in Figure 5 seem to suggest discrimination in single-nucleotide level, but it remains to be seen whether this holds up in more complex sample material. Furthermore, I have some reservations whether this is an inherent property of SAHARA or just of the wild-type Cas12 itself (see my comments below). The authors should at least discuss these potential limitations in the discussion.

Response: We thank the reviewer for the feedback. We understand the potential limitation of our technology with respect to detecting RNA in more complex sample material. We have added a detailed commentary regarding this to our discussion. To investigate this further, we spiked the 86-nt long HCV-RNA that we had previously tested in Fig. 4 to a matrix consisting of extracted nucleic acids from healthy human serum samples. We compared the detection efficiency of the target RNA spiked in serum vs. target RNA in water. We used ErCas12a for this experiment.

Our results indicated that SAHARA was able to robustly detect the target RNA in both water as well as serum matrix. We observed that compared to water, there was a slight increase in the background activity for the RNA spiked in serum. Now this increase in background activity might be attributed to the non-specific cleavage of reporters in the serum or because of the interactions with similar RNA sequences in the serum matrix, as the reviewer hypothesized. However, we also observed that the on-target activity increased proportionally. Therefore, the fold-change of target detection in both water and serum remained the same, even though the overall fluorescence generated is higher in serum. This suggests that the serum only enhanced the background cleavage activity without affecting the specificity of the SAHARA.

In a diagnostic setting, depending on the kind of matrix being tested (blood, serum, saliva, urine etc.) we recommend using a healthy matrix as a control to account for background activity arising from non-specific interactions within the matrix, and then incorporating statistical methods to detect the presence or absence of target RNA based on the differences in the fluorescence intensity with respect to the control. We have added this recommendation in the discussion.

Fig S15: Target HCV RNA was spiked in a matrix consisting of either water or nucleic acid extract from healthy serum sample. The detection efficiency of target RNA spiked in water or serum was determined with SAHARA. (a) Plot representing RFU of target RNA detection in water or serum at t=60 min. (b) Plot representing fold change of fluorescence from target RNA sample with respect to the fluorescence from the no target control (NTC) in water and serum.

R2.2 The authors quantify the trans-cleavage activity as either fold-change, delta RFU or raw RFU in different parts of the manuscript. The authors should at least elaborate on the reasoning behind this, and why not being consistent with this throughout the manuscript, as it doesn't allow to make comparisons between the results.

Response: We thank the reviewer for the feedback. We initially used fold-change with respect to the no target control (NTC) to represent our data, as is standard for the field as well as to be consistent with previous publications from our lab². This is why all our initial experiments in Fig. 1 and most experiments in Fig. 2 are represented as fold change. However, when we started doing 'split-activator' type of experiments with SAHARA, which consists of an artificial S12-DNA and well as the target DNA or RNA, we noticed that the S12-DNA by itself would often initiate a small amount of background collateral cleavage activity of Cas12a, even without the PAM-distal target RNA or DNA being present in the reaction. This is especially true of AsCas12a, which has the highest amount of background collateral cleavage activity as is evident from Fig. 3c as well as Fig. 6i. However, this information gets hidden if the data is represented as fold-change. Therefore, to better represent this background activity, we decided to switch to using raw RFU or delta RFU to show the data, instead of fold change.

R2.3 Page 8 and Figures 2 and S3: The authors observed that PAM-containing dsDNA activators bolster trans-cleavage activity. To study the influence of the PAM, the authors conducted a number of experiments with different activator combinations, presented in Figure S3. However, the combinations tested are not adequate to address this. Firstly (judging from the schematics), the Pd substrates have the PAM at the wrong position (hence, it can never function as a PAM). Secondly, the authors compared a (Pp) PAM-containing substrate with a shorter substrate lacking nucleotides at the position of where the PAM would have been. A better comparison would be to test the same length substrate where the PAM is replaced by a non-PAM sequence.

Response: We appreciate the reviewer's feedback. We agree with the reviewer's comments about the position of the PAM sequence at the Pd as well as the length of the PAM containing substrate at the Pp for the data presented in Fig. S3. Due to the multiple valid concerns that the reviewer pointed out for Fig. S3, we have decided to remove this figure from the final version of this manuscript. We believe that we have adequately addressed the question about the requirement of a PAM sequence in the S12-DNA later in the manuscript in Fig. 6b-d, and therefore feel that the data presented in Fig. S3 does not add much to the overall narrative of this work.

R2.4.1 Page 14: "We observed that mutations in positions M01-M06 significantly decreased the SAHARA-mediated trans-cleavage activity"

1. Since these mutations are not part of the seed sequence, this is quite unexpected. Is this consistent with other literature (especially since this effect also seems to be existent for the WT-Cas12)?

We thank the reviewer for the feedback.

Response: For the wild-type (WT) CRISPR-Cas12a, we have observed a similar reduction in the activity for mutations at the identical positions in one of our earlier publications² (Nguyen et al., *Nat. Comms.*, 2020, Figs. 4b,f, S13-S18). Compared to the wild-type target, the trans-cleavage activity for the mutant targets was reduced to half or less in that study.

For SAHARA, due to the nature of the split-activator system, it makes sense that the mutations closer to the interface of the two targets will have a profound effect on the trans-cleavage activity. When you look at Figs. 1e-g, split activator combinations that are not perfectly matched (for e.g., Pp 12-nt + Pd 6-nt, a combination that will have a 2-nt gap in between), do not show any trans-cleavage activity. We think the decrease in the trans-cleavage activity for SAHARA for mutations at M01-M06 (now renamed as M13-M18) might be due to the mismatch base-pairing at the interface of the two targets.

We would also like to highlight that while both WT CRISPR-Cas12a as well as SAHARA displayed reduced activities for the mutations, the effect was much more pronounced for SAHARA when compared to WT CRISPR-Cas12a for M01 to M06 (now renamed as M13-M18). This is the basis for our claim that SAHARA is more specific to point mutations at positions M13-M18.

R2.4.2 Likewise, the authors also mention that "Mutations in positions beyond M06 did not decrease the trans-cleavage activity for either Cas12a ortholog", however, even the most seed distal M012 mutation appears to impact trans-cleavage activity for both WT as SAHARA (Figure 5c).

Response: We thank the reviewer for the feedback. We want to clarify that we are merely comparing the decrease in the trans-cleavage activity of SAHARA as compared to the WT CRISPR-Cas12a for point

mutations along the target. As per Fig. 5c only mutations in the positions M01-M06 (now renamed as M13-M18), show a decrease in the trans-cleavage for SAHARA as compared to the WT CRISPR-Cas12a. For mutations at positions beyond M06, SAHARA becomes equivalent (or worse) as compared to the WT CRISPR-Cas12a for point mutations. To clarify, we have add the following sentence in the text:

We used the mutant activators to compare the trans-cleavage activity of SAHARA with the activity of WT CRISPR-Cas12a for point mutations along the target.

R2.4.3. The authors labeled this section “SAHARA improves the specificity of target detection”, but the results seem to be quite like the wild-type Cas12 setup (and not consistent across the different orthologues tested). Perhaps a better conclusion would be to say that SAHARA has regained its sequence specificity (when compared to the wild-type Cas12 setup)?

Response: We thank the reviewer for the feedback. According to the presented data, for point mutations along positions M01-M06 (renamed to M13-M18), SAHARA seems to have ~1.8x – 37x lower activity as compared to the WT CRISPR-Cas12a system. This seems to suggest that SAHARA is extremely sensitive towards point mutations near the interface of the two activators in the split-activator system (please refer to the next comment R2.4.4 for an updated Fig. 5). We agree that labeling this section as ‘SAHARA improves specificity of target detection’ might be misleading. Therefore, we have changed our title to ‘SAHARA displays improved specificity towards detection of point mutations.’

A recent paper that was published after our initial submission also shows a similar single-nucleotide specificity using split-activators³.

R2.4.4 The authors should consider switching their nucleotide numbering to a more conventional standard (where position 1 is the first base of the protospacer, base-pairing to the 5’ most base of the spacer region of the crRNA, see for example Zetsche et al, doi.org/10.1016/j.cell.2015.09.038)

Response: We thank the reviewer for the feedback. We have changed the nucleotide numbering as per the reviewer’s recommendation.

Fig. 5: Specificity of SAHARA towards single point mutations in target a. Schematic of WT vs. SAHARA CRISPR-Cas12a systems for the detection of a target nucleic acid. **b.** ssDNA activators were designed with point mutations across the length of the activator. GFP-activator mutants were designed for a WT CRISPR activator (24-nt) and a SAHARA split activator system (12-nt +12-nt). The mutation location is identified by 'M' following the nucleotide number where the base has been changed to guanine (3' to 5' direction). **c-e.** Comparison of fold changes for the *in vitro trans*-cleavage assay between WT and SAHARA activator mutants normalized to the WT activator for Cas12a orthologs (c: LbCas12a, d: AsCas12a, and e: ErCas12a). Comparison of RFU values at t=60 min for the *in vitro trans*-cleavage assay between WT and SAHARA. Statistical analysis was performed using a two-tailed t-test where ns = not significant with $p > 0.05$, and the asterisks (* $P \leq 0.05$, ** $P \leq 0.01$, *** $P \leq 0.001$, **** $P \leq 0.0001$) denote significant differences.

R2.5 The discussion reads more like a repetition of the result section. In this reviewer's opinion, this space could better be used for other matters, such as the benefits, limitations, and outlook of their SAHARA technology, how it compares to the wealth of other Cas12-based diagnostic platforms that have been published before, etc.

Response: Thank you for the recommendation. We have modified the discussion section of the manuscript accordingly. Reproduced below:

“While CRISPR-Cas systems have been studied for over a decade, *trans*-cleaving Cas enzymes such as Cas12 and Cas13 are relatively new. Despite their novelty, tremendous progress is being made towards studying and understanding their underlying mechanism, especially due to their utility in molecular diagnostics. Cas12a, in particular, has been extensively studied and widely used in DNA-based diagnostic platforms. However, its use has been limited to DNA detection, as it does not naturally tolerate RNA substrates. In this work we have discovered that Cas12a's tolerance for RNA substrates is position-dependent on the crRNA. By binding RNA substrates at the PAM-distal end of the crRNA and providing a DNA substrate at the PAM-proximal end, the *trans*-cleavage activity of Cas12a can be activated.

In contrast to Cas9, which uses two different active sites to generate a blunt double-stranded DNA break⁴, Cas12a uses a single active site to make staggered cuts on the two strands of a dsDNA⁵⁻⁷. After cleaving the target DNA, Cas12a releases the PAM-distal cleavage product while retaining the PAM-proximal cleavage product bound to the crRNA^{8,9}. This maintains Cas12a in a catalytically competent state, in which the active site of RuvC remains exposed to the solvent which then leads to *trans*-cleavage of neighboring single-stranded DNA molecules in a nonspecific manner.

It has been shown that crRNA-target DNA hybrids of length 14-nt or less do not trigger any *cis*- or *trans*-cleavage activity, and a crRNA-DNA hybrid of at least 17-nt is crucial for stable Cas12a binding and cleavage. These observations suggest that the interaction of Cas12a with crRNA-target DNA hybrid at positions 14-17 nt is critical for initiating cleavage^{8,10,11}. Truncating the length of single-stranded DNA activators that bind to the crRNA significantly reduces the *trans*-cleavage activity of Cas12a. However, we found that using two truncated activators, each binding to a different region of the crRNA, partially restores the lost activity.

Further experiments revealed that the PAM-proximal seed region of the crRNA exclusively tolerates DNA for initiating *trans*-cleavage, while the PAM-distal end can tolerate both RNA and DNA, except in the case of AsCas12a which interestingly tolerated RNA even at the Pp region. By designing long crRNAs with specific regions complementary to PAM-containing dsDNA and the target of interest, we developed a split-activator-based method called SAHARA for RNA detection with Cas12a. SAHARA enabled robust RNA detection activity for various lengths of RNA, with ErCas12a displaying the strongest activity for long RNA sequences.

We applied SAHARA for the detection of clinically relevant targets such as Hepatitis C virus (HCV) and miRNA-155. We found that the secondary structure of the target played an important role in the RNA detection activity of Cas12a. Increased secondary structure in the RNA target makes it more inaccessible to bind to the crRNA and reduces the activity. We also observed that pooling together multiple crRNAs targeting different regions of the activator enhanced the detection capability, as previously reported for Cas13. Applying this, we were able to detect picomolar levels of both miRNA-155 as well as HCV with SAHARA.

While SAHARA works well with synthetic RNA targets, a key limitation of our study is the lack of validation with respect to detecting RNA samples with more complex material in a clinical setting. A

potential drawback of SAHARA is its specificity in more complex samples. Since SAHARA is designed to only bind to 12-nt of the target RNA, the odds of finding similar RNA sequences in a complex material are high, potentially resulting in false-positives. Upon testing detection of a HCV target spiked in nucleic acids extracted from healthy serum samples, we observed an increase in the background activity resulting from SAHARA as compared to detection of the same sample in water. However, we also observed that the on-target activity increased proportionally.

In a diagnostic setting, depending on the kind of matrix being tested (for e.g. blood, serum, saliva, urine etc.) we recommend using a healthy matrix as a control to account for background activity arising from non-specific interactions within the matrix, and then making use of statistical methods to detect the presence or absence of target RNA based on the differences in the fluorescence intensity with respect to the control.

Developing a robust, clinically applicable, diagnostic test with SAHARA will require additional efforts, especially with regards to improving not just the specificity but also the sensitivity of detection. SAHARA in its current form is limited to only picomolar levels of detection without amplification, which may not be good enough for detection of clinical samples. However, a promising outlook lies in combining SAHARA with the plethora of amplification-free detection techniques that can be game-changing for the field of Cas12a-based diagnostics.

MINOR COMMENTS (R2.6)

R2.6.1 Page 3: “or a PAM containing dsDNA”
Consider changing this to “PAM-containing”

Response: Thank you. We have made the changes.

R2.6.2 Page 3: “Notably, SAHARA is Mg²⁺ concentration- and pH-dependent”
Not sure whether this is worth mentioning in the abstract

Response: Thank you. We have removed this sentence.

R2.6.3 Page 4: “Cas (CRISPR-associated) proteins are RNA-guided endonucleases...”
This is not an accurate description of Cas proteins (i.e. not all Cas proteins are nucleases). Please adjust accordingly.

Response: Thank you. That is a valid point. We have changed the sentence to the following:

“CRISPR/Cas (CRISPR-associated) systems contain RNA-guided endonucleases that when complexed with the CRISPR RNAs (crRNAs) can enable the cleavage of nucleic acids that are complementary to the crRNA sequence.”

R2.6.4 Page 4: “Cas12a is a class II,”
Please change this to “class 2”

Response: Thank you. We have made the changes.

R2.6.5 Page 4: “...there are no reports of targeted RNA cleavage by Cas12a.”
Please delete or rephrase, as this contradicts with the next sentence.

Response: Thank you. In the next sentence we are talking about the recently discovered Cas12a2 system, which is a class of enzyme distinct from Cas12a¹². We just want to highlight that Cas12a2, which is an RNA-targeting enzyme, has been reported to sometimes co-occur with Cas12a and is able to interchange guides with Cas12a. However, the statement about no reports of RNA targeting by Cas12a itself still holds true.

R2.6.6 Page 5: “...also cleave RNA through the addition of a PAMmer sequence.”
Given the relevance of this concept for the findings in this manuscript, it would be helpful (especially for a non-expert reader) if the authors could shortly explain what a PAMmer sequence is.

Response: Thank you for the suggestion. We have added the following sentence:

“.. the addition of a PAMmer sequence, a PAM-containing DNA oligonucleotide which is annealed to the target ssRNA to initiate cleavage.”

R2.6.7 Results

Page 7: “...and is completely lost for activators less than 12-nt in length”
Looking at figures 1-d, perhaps the authors meant to say “less than 14-nt in length” here?

Response: We have changed this sentence to “less than or equal to 14-nt in length” since AsCas12a is able to detect a 14-nt target.

R2.6.8 Page 7: “...and mimicking a full-length target, would be able to regain the lost trans-cleavage activity.”

This is somewhat confusing, why “lost”? My guess is that the authors are referring to the lack of trans-cleavage activity when using single ≤ 14 -nt DNA activators; this should be explained better.

Response: Thank you for the suggestion, we have reworded this sentence as follows:

Next, we tested to check if the simultaneous addition of two truncated activators, each binding to different regions of the crRNA and together mimicking a full-length target, would be able to regain the diminished trans-cleavage activity observed for shorter (≤ 14 -nt) activators

R2.6.9 Page 9: “...that both ssDNA and ssRNA had significantly greater activity with the EN crRNA for LbCas12a” Did the authors apply any statistical test to substantiate this claim?

Response: Thank you for the feedback. We have added statistical analysis to this figure.

Fig. 2 Split-activator detection of ssDNA, dsDNA, and RNA substrates by Cas12a: a. Schematic representation of Cas12a activated by combinations of ssDNA (red), dsDNA (orange), and RNA (blue) in the PAM proximal and PAM distal regions. b-d Heat maps representing the fold changes of *in vitro* trans-cleavage assay (n=3) with Cas12a orthologs for the combinatorial schemes seen in (a). e-h Comparison of

the WT crRNA and ENHANCE crRNA for *in vitro trans*-cleavage assay split activators. Note, ssDNA and ssRNA substrates were used as targets in the Pd region while dsDNA was supplied in the Pp. Reactions were incubated for 60 min at 37°C. Error bars represent SD (n=3). The reactions contained 25 nM of each truncated activator, 60 nM Cas12a, and 120 nM crRNA.

R2.6.10 Page 9: “This is consistent with previous observations with ENHANCE.”
Please provide a reference.

Response: Thanks for pointing it out. We have added a reference.

R2.6.11 Page 9 : “irrespective of the crRNA design being used. (Fig. 2f-h).”
Please delete the first full stop.

Response: Thanks for pointing it out. We have deleted the full stop.

R2.6.12 Page 10: “...somewhere between the seed region and the 3’-end of the crRNA protospacer”
Perhaps the authors meant spacer instead of protospacer here? Consider rephrasing this to “3’ spacer-end of the crRNA”. Please adjust “crRNA protospacer” on page 11 accordingly.

Response: Thanks. We have made both changes.

R2.6.13 Consider moving the paragraph “While a fully complementary RNA activator...can detect both RNA and DNA.” (Testing the limits of RNA detection) after the text describing the results of Figure 2 (showing that RNA can be detected).

Response: Thank you for the suggestion. We have moved this paragraph after the results of Fig 2.

R2.6.14 Page 11: The sentence “...and discovered that most metal ions including severely inhibited the trans-cleavage activity of SAHARA” seems to be incomplete (including what?).

Response: Thanks for pointing this out. We have changed the sentence as follows:
“discovered that most cations severely inhibited the trans-cleavage activity of SAHARA (Fig. S9).”

R2.6.15 Page 11: “...consistent with the literature.”
Please cite the literature that is being referred to here.

Response: Thanks for pointing this out. We have added a reference to our previous study where we tested the effect of different metal ions on the trans-cleavage activity of Cas12a² (Nguyen et al., *Nat. Comms.*, 2020).

R2.6.16 Page 12: “targeting it at either 12-nt at the 5’-end or 11-nt at the 3’-end, keeping the S12-binding region constant”
The authors should refer to Figure 4d here.

Response: Thank you. We have added a reference.

R2.6.17 Page 12: “SAHARA also displayed a higher activity with pooled crRNAs for both miRNA-155 and

HCV targets.”

The authors should refer to the corresponding figures here.

Response: Thank you. We have added the reference.

R2.6.18 Page 14: “TTTA is one of the canonical PAM for Cas12a”

The authors might want to be a bit more clear on what the tolerated PAMs for the different Cas12a orthologues are. For instance, ErCas12 has been reported to recognize YTTN PAMs (10.1089/crispr.2019.0026), potentially explaining why “Er Cas12a was able to detect RNA even with an VVVN PAM”.

Response: Thank you for pointing this out, we have reworded this section appropriately. Please see below:

“TTTA is one of the canonical PAM for Lb and As Cas12a, while Er is canonically known to have a preference for YTTN PAM¹³, AAAT is the PAM-complement (cPAM) sequence, and VVVN encompasses all the PAM sequences that are not tolerated by Cas12a. We tested the detection of a short 20-nt RNA with these S12 activators. As expected, the TTTA-PAM S12 was able to mediate RNA detection with all three Cas12a orthologs (Fig. 6b-d). Upon changing the PAM from TTTA to AAAT or VVVN, the *trans*-cleavage activity was completely diminished for Lb and As. Interestingly, Er Cas12a was able to detect RNA even an AAAT PAM, hinting at a broader PAM tolerance for *trans*-cleavage than previously reported.”

R2.6.19 In this reviewer’s opinion, the “S12 DNA can be repurposed as a switch to control the trans-cleavage activity of SAHARA” is a bit unnecessary, as many of the outcomes are to be expected, namely that a) non-canonical PAMs fail to stimulate trans-cleavage activity, b) that GC-content has no influence (why would it?) and c) that S12 is critical for the initiation of trans-cleavage activity (which can already be concluded from the results in Figure 3).

Response: We thank the reviewer for their feedback. Even though the results related to PAM-requirement are to be expected for target DNA cleavage with Cas12a, since we are testing a new RNA-targeting mechanism with Cas12a we thought it would be best to test it to be thorough in our characterization. The GC content of the spacer region has also been previously reported to affect the activity of Cas12a^{14,15}.

Finally, even though we first highlight the requirement of S12 DNA to be critical for SAHARA’s activity in Fig. 3, in this section we have expanded this result further by testing different concentrations of S12 DNA and showing that stoichiometrically very low amounts of S12 DNA can be used to turn on SAHARA.

R2.6.20 Page 15: “we used an ssDNA target A for crRNA-a while targets B and C were ssRNAs”

Please be consistent when referring to the targets (in the text they are called targets A,B,C, whereas in Figure 7 they are called targets 1,2,3). Also indicate in the figure whether a target is RNA or DNA.

Response: Thank you for pointing this out. We have changed the Fig. 7 to correctly indicate targets A, B and C.

R2.6.21 Page 15: "...in the presence or absence of the corresponding S12. We also used a control wherein no S12 sequence was supplied to the crRNA-Cas-target mix."

This is ambiguous, is there a difference between “absence of ...S12” and “no S12 sequence”??

Response: Thanks for pointing this out. We have deleted “We also used a control wherein no S12 sequence was supplied to the crRNA-Cas-target mix.”

R2.6.22 There are no references to Figures 7a-e throughout the whole manuscript!

Response: We appreciate the reviewer for noticing this. We have added the necessary references.

R2.6.23 Page 16: “We also used a FAM-containing DNA reporter and a HEX-containing RNA reporter to distinguish the signals of SAHARA and Cas13b.” seems to be repeated just a few lines down:

“Furthermore, the

trans-cleavage signal obtained from SAHARA and Cas13b could be distinguished from each other by using orthogonal fluorescent dyes such as FAM and HEX on different types of reporter molecules”

Response: Thank you. We have deleted “We also used a FAM-containing DNA reporter and a HEX-containing RNA reporter to distinguish the signals of SAHARA and Cas13b.”

R2.6.24 Page 17: “While CRISPR-Cas12a systems have been studied for almost a decade, trans-cleaving Cas enzymes such as Cas12 and Cas13 are relatively new”

CRISPR-Cas12a systems have been studied well over a decade, starting in 2007:

(doi.org/10.1126/science.1138140) and trans-cleaving activity has already been shown in 2018

(doi.org/10.1126/science.aar6245)

Response: Thanks for the feedback. We have modified the sentence to “...over a decade...”

R2.6.25 Page 19: “all three cas12a”

Change to “all three Cas12a”

Response: Thank you. We have made the change.

Figures and Legends

R2.6.26 Figure 1b-d: it is unclear how the Pp and Pd DNA activators are truncated and designed, are these all 3' truncations (or 3' truncations for Pd and 5' for Pp?), what regions are part of the protospacer and/or seed sequence, where is the transition between Pp and Pd? etc.

Response: Thank you for the feedback, we have modified the schematic in Fig. 1 to indicate how the Pp and Pd activators bind to the crRNA. The Pp activators all have 3'-end truncations while the Pd activators have 5'-end truncations. We have added a schematic in the supplementary material to demonstrate how the different length activators bind to the Pp and Pd.

Fig. 1: Cas12a orthologs tolerate short ssDNA activators (6-12 nt) when added in combination. Schematic representation of a crRNA-Cas12a complex performing *trans*-cleavage of ssDNA reporters following the recognition of two split-activators. **b-d.** Fold change with normalized to No Target Control (NTC) at t=60 minutes of *in vitro trans*-cleavage assay with Cas12a orthologs (red = LbCas12a, green = AsCas12a, orange = ErCas12a) activated by individual truncated ssDNA activators of length 6-20 nt. Statistical comparisons are made for a combination of both Pp and Pd data sets between the different lengths of the targets. Statistical analysis was performed using a two-tailed t-test where ns = not significant with $p > 0.05$, and the asterisks (* $p \leq 0.05$, ** $p \leq 0.01$, *** $p \leq 0.001$, and **** $p \leq 0.0001$) denote significant differences. **e-g.** Heat maps representing fold change with respect to NTC at t=60 minutes of an *in vitro trans*-cleavage assay activated by combinations of truncated ssDNA activators of different lengths ranging from 6-14 nt in the Pp and Pd regions. The reactions contained 25 nM truncated ssDNA GFP-activators, 60 nM Cas12a, and 120 nM crGFP and were incubated for 60 min at 37°C. The NTC represents the condition when neither the Pp nor the Pd activator is present in the reaction. Error bars represent SD (n=3).

Fig. S3: Schematic depicting how truncated ssDNA activators of different length bind to the PAM-proximal (Pp) and PAM-distal (Pd) regions of the crRNA.

R2.6.27 In addition, the authors should explain what is meant by “NTC”. Without this information, the data is very hard to interpret. The authors should clarify this the figure and text.

Response: The NTC represents the ‘No Target Control’. This represents the control wherein neither of the split-activator targets are included in the reaction. We have added a definition for NTC in the figure caption for Fig.1.

R2.6.28 Figure 1e-g. It is unclear why target combinations with a total length longer than 20 nt (e.g. 14-nt Pp + 8-nt Pd) seem to perform worse compared to their 20-nt counterparts. Do the authors have an explanation for this? Similar to my previous comment, it is also unclear how the >20-nt version are extended (at the 5' for the Pd ones?).

Response: Thank you for the question.

The Pp and Pd targets of length > 20-nt are extended as shown in the new Fig.1 schematic. More specifically, the Pp targets are extended by adding nucleotides in the direction moving towards the 3'-end of the crRNA while the Pd targets are extended by adding nucleotides moving towards the 5'-end of the crRNA. See below for the schematic.

Response: Target combinations with length > 20-nt will therefore, overlap in the center of the guide RNA binding region. These overlapping regions might sterically inhibit binding of the target near the split-activator boundary due to excess nucleotides. We think this is the reason why activator combinations of total length > 20-nt perform worse than their 20-nt counterparts.

R2.6.29 Figure legend 1b-d: “Fold change at t=60 minutes of in vitro trans-cleavage assay...”
Please revise (what does fold change of an assay mean, fold change in what, compared to what?)

Response: Thank you for the feedback, we have added a definition for the fold change as “fold change normalized to NTC” in the figure caption.

R2.6.30 Figure 3a: please indicate in the legend that the red part of the long RNA represents the target sequence.

Response: Thank you for the suggestion, we have added the following sentence to Fig. 3a: “The orange section of the long RNA activator corresponds to the target sequence.”

R2.6.31 Figures 3e-g: This is quite confusing. 1) What RNA was used as a target and why not both as in panels b-d? 2) Is “SR-12” the same condition as “SAHARA” (in panels b-d)? If so, why not label them the same? 3) If both the “SR-12” and “WT” conditions are the same as in panels b-d, why not add the scrambled data there? It seems a bit excessive to spend 3 panels on data showing that the scrambled guide RNA doesn’t result in trans-cleavage activity.

Response: Thank you for the question.

- 1) Only the long RNA was used for the experiments done in the Fig. panels 3e-g, we have added this information in the figure caption. We did not use both targets to reduce the complexity of the experiment.
- 2) Yes, it is the same as SAHARA, we changed the figure label to make it consistent.
- 3) We did these two experiments separately; therefore, we made separate charts to represent them.

R2.6.32 Figure S6: What RNA target was used? More importantly, since the authors wanted to test whether longer (>20 nt) crRNA perform better, why was the canonical 20 nt crRNA not included as a benchmark?

Response: Thanks for pointing that out. For this experiment, we used the HCV RNA as the target. We have added this information in the figure caption. We only tested crRNA spacer lengths between 24-28 nt since below these lengths (for e.g. 20 nt) the target RNA binding region will be very short (only 10-nt) and therefore highly non-specific for any practical use.

R2.6.33 Figure S7: what does “w.r.t.” stand for?

Response: w.r.t stands for ‘with respect to’. We have expanded the abbreviation in our captions.

R2.6.34 Figure 5: It is a bit confusing that the orientation of the Cas12 schematics has been mirrored when compared to the previous schematics, which concomitantly forced the authors to depict sequences in the 3' to 5' orientation. Why not keep it consistent? Also, please indicate which Cas12a orthologue was used in the figure.

Response: We thank the reviewer’s comment. As suggested, we have decided to keep the Cas12a schematics in Figure 5 consistent with the other figures. We have changed the schematic for all figures depicting crRNA and Cas12a such that the pseudoknot of the crRNA is on the left-hand side. We have also indicated which bar graphs are with what Cas12a orthologues.

REVIEWER #3 (REMARKS TO THE AUTHOR): R3

Rananaware et. al. describes the ability of three orthologs of CRISPR-Cas12a to detect single-stranded DNA and RNA targets when a small 12 nt duplex containing a PAM and seed is also present. The method described by the authors (SAHARA) expands the utility of Cas12a to detect RNA without the need for reverse transcription steps. The authors also demonstrate the potential of SAHARA in multiplexed detection of both DNA and RNA targets in pooled samples. This work reveals an interesting functionality of Cas12a that was not previously realized that could be used to expand Cas12a-based diagnostics. However, more needs to be done to establish SAHARA as a viable RNA diagnostic tool.

R3.1 My primary concern is that the authors did not validate RNA diagnostic activity in the context of what would be reflected in a true disease diagnosis situation. For example, the HCV RNA used in this study was only 86 nt long. The validation experiment should be done with the entire HCV genome and with background RNA that would normally be present in a patient sample. Indeed, AsCas12a was not able to distinguish long RNAs from background in a previous experiment, suggesting the length of the RNA can influence the ability to detect RNA. Additionally, the authors did not indicate which ortholog of Cas12a was used to do the validation tests.

Response: We thank the reviewer for this feedback. Please see our response below.

Validation in a true diagnostic setting:

We understand the limitation of our study with respect to validation of our technology in a true disease setting. We would like to highlight that our primary motive with this manuscript was not to develop a robust diagnostic test but rather to demonstrate this unusual ‘split-activator’ property of Cas12a enzymes that can potentially be harnessed to directly perform RNA detection with a strictly DNA-tolerating enzyme, that would not have been possible before. However, inspired by the reviewer’s concern we performed two sets of experiments as discussed below that demonstrates the power of SAHARA for potential clinical applications.

Detection in the presence of background RNA:

To address reviewer’s concern, we investigated the ability of SAHARA for detection in a more complex environment. We took the 86-nt long HCV-RNA that we had previously tested in Fig. 4 and spiked it in a matrix consisting of extracted nucleic acids from healthy human serum samples. We compared the detection efficiency of the target RNA spiked in serum vs. target RNA in water. We used ErCas12a for this experiment.

Our results were exciting and indicated that SAHARA was able to robustly detect the target RNA in both water as well as serum matrix. We observed that compared to water, there was a slight increase in the background activity for the RNA spiked in serum. We think this increase in background activity might be on account of non-specific interactions with similar RNA sequences in the serum matrix, as the reviewer hypothesized. However, we also observed that the on-target activity increased proportionally. Therefore, the fold-change of target detection in both water and serum remained the same, even though the overall fluorescence generated is higher in serum.

In a diagnostic setting, depending on the kind of matrix being tested (blood, serum, saliva, urine etc.) we recommend using a healthy matrix as a control to account for background activity arising from non-specific interactions within the matrix, and then making use of statistical methods to detect the presence or absence of target RNA based on the differences in the fluorescence intensity with respect to the control. We have added this recommendation in the discussion.

Fig S15: Target HCV RNA was spiked in a matrix consisting of either water or nucleic acid extract from healthy serum sample. The detection efficiency of target RNA spiked in water or serum was determined with SAHARA. (a) Plot representing RFU of target RNA detection in water or serum at t=60 min. (b) Plot representing fold change of fluorescence from target RNA sample with respect to the fluorescence from the No target control (NTC) in water and serum.

Detection of longer RNA:

We have previously shown the detection of a longer (~730-nt) with SAHARA (Fig. 3b-d). However, the target used in that experiment was GFP. To demonstrate the same with a more clinically relevant RNA target, we ordered the Quantitative Synthetic RNA from Hepatitis C virus from ATCC (Cat# VR-3233SD) that consists of fragments from 5'UTR and X-tail region, thus being a more complex sample as compared to the previously used 86-nt RNA. Initially, we found the detection of this RNA target with our typical protocol to be challenging due to the low concentration of this target RNA (10^5 - 10^6 copies/ μ L). However, we were able to detect this RNA upon increasing the concentration of crRNA, Cas12a, and S12 to 250 nM, 125 nM, and 105 nM respectively (Fig. S14). We used ErCas12a for this experiment, since this ortholog works the best for detection of longer RNA sequences.

Fig S14: Quantitative Synthetic RNA from Hepatitis C Virus purchased from ATCC (HCV-ATCC) was tested for detection with ErCas12a based SAHARA. 250 nM of a crRNA pool consisting of HCV-Tail, HCV-Mid and HCV-Head; 125 nM Cas12a; 500 nM FQ and 105 nM S12-DNA was used in the reaction. Data represents an increase in RFU vs time for a time-course of 1 hr. Error bars represent S.D. (n=3).

We understand to take this technology and develop it into a robust diagnostic test will require additional efforts, especially with regards to improving the sensitivity of detection, since SAHARA in its current form is limited to only picomolar levels of detection without amplification, which although impressive may not good enough for detection of clinical samples. However, we envision that combining SAHARA with the plethora of amplification-free detection techniques that have recently emerged for CRISPR-Cas12a and CRISPR-Cas13a based platforms can be game-changing for the field of Cas12a based diagnostics. We have also added a detailed regarding the limitations of this technology in the discussion.

R3.2 The work performed to evaluate specificity compared point mutants of a single stranded DNA target to a completely complementary target. Although the data revealed some differences in mismatch tolerance between the SAHARA and WT the authors should evaluate how mutations in RNA targets influence the diagnostic activities as ssDNA substrates will not be available in-patient diagnostic samples. The authors should also evaluate specificity in the presence of competing non-target RNA and the RNA targets used should be of a length similar to what would be observed in a natural sample instead of the exact complement to the split guide.

Response: We appreciate the reviewer’s feedback. For this study we aimed to evaluate the specificity of point mutation detection with SAHARA and compare it to the wild-type CRISPR-Cas12a system. Our data revealed that compared to the WT CRISPR-Cas12a system, SAHARA is more specific towards point mutations when closer to the boundary of the two split-activators. We chose to perform the SAHARA specificity study with mutated ssDNA targets because we could compare it with the WT CRISPR-Cas12a, full-length ssDNA targets as shown in the Fig. 5. Since the WT CRISPR-Cas12a cannot detect full length RNA targets, there will be no baseline for comparison.

As the reviewer indicated, we have now performed the testing of RNA targets in water vs. extracted serum that contains a complex mixture of competing non-target RNAs. We observed that compared to water, there was a slight increase in the background activity for the RNA spiked in serum. We think this increase in background activity might be on account of non-specific interactions with similar RNA sequences in the serum matrix, as the reviewer hypothesized. However, we also observed that the on-target activity increased proportionally. Therefore, the fold-change of target detection in both water and serum remained the same, even though the overall fluorescence generated is higher in serum.

Fig S15: Target HCV RNA was spiked in a matrix consisting of either water or nucleic acid extract from healthy serum sample. The detection efficiency of target RNA spiked in water or serum was determined with SAHARA. (a) Plot representing RFU of target RNA detection in water or serum at t=60 min. (b) Plot representing fold change of fluorescence from target RNA sample with respect to the fluorescence from the No target control (NTC) in water and serum.

Minor comments (R3.3)

R3.3.1 The authors should consider presenting the Cas12a with the crRNA hairpin and PAM-binding region on the left side of their Cas12a diagrams as shown in figure 5. The inverse of this depiction with the 3' end of the guide RNA on the left, shown in figures 1.,2.,3., 6., and 7. is opposite of how Cas12a is usually depicted schematically in the field. This inverse orientation becomes particularly problematic in figure 3, when the PAM sequence is indicated in panel A. The canonical 5'-TTTA-3' PAM indicates the PAM sequence reading left to right, but the way the PAM is depicted it should be inverted to 3'-ATTT-5' to match with the orientation of the PAM in the diagram.

Additionally, crystal structures indicate the PAM remains double stranded when bound by Cas12a. However, several figures depict Cas12a bound to a duplex with an unwound PAM. A more accurate diagram of Cas12a bound to a PAM containing duplex should be diagrammed instead.

Response: Thank you for the suggestion. This makes sense. We have modified our figures accordingly.

R3.3.2 To better reflect the design of the dsDNA duplex, in the abstract consider replacing the phrase “a PAM containing dsDNA to the seed region” with “a dsDNA spanning PAM and seed regions.”

Response: Thank you for the suggestion. We have modified the abstract accordingly.

R3.3.3 The statement “Cas (CRISPR-associated) proteins are RNA-guided endonucleases that when complexed with the CRISPR RNAs” is somewhat inaccurate as (i.) not all Cas proteins are endonucleases, and (ii) not all Cas proteins complex with crRNA.

Response: Thank you. That is a valid point. We have changed the sentence to the following:

“CRISPR/Cas (CRISPR-associated) systems contain RNA-guided endonucleases that when complexed with

the CRISPR RNAs (crRNAs) can enable the cleavage of nucleic acids that are complementary to the crRNA sequence.”

R3.3.4 Usually, Class 2 CRISPR systems are indicated with a Hindu-Arabic numeral 2 instead of roman numeral II.

Response: Thank you for the suggestion. We have made this change.

R3.3.5 Can the authors address whether split targets (both DNA and RNA targets) are still cleaved in cis when they are long enough to fit into the RuvC endonuclease site?

Response: Thank you for the question. We think cis-cleavage might not be necessary for detecting single-stranded RNA targets since compared to dsDNA targets they can fold more easily making the RuvC endonuclease solvent accessible for trans-cleavage of other ssDNA targets. This is something that we are still actively investigating. Upon testing the cis-cleavage for five different RNA targets, we observed that only 1/5 targets was being cis-cleaved (even though trans-cleavage was active for all five). We do not yet have a conclusive understanding of whether cis cleavage is occurring or not, therefore we have not included the data, especially since the major focus of this manuscript is on the trans-cleavage activity.

R3.3.6 The placement of significance brackets in figure 1.b-d makes it unclear which data are being compared. It seems as if comparisons are between Pd data sets. Is this correct? It should be made clearer in the figure legend what data sets are being compared.

Response: Thank you for the question, the significance brackets are plotted for a combination of both Pp and Pd data sets for the for the different lengths of the targets. We have added a sentence reflecting that in the figure caption:

Pp and Pd data sets of varying target lengths are combined for a comprehensive statistical analysis.

R3.3.7 Can the authors address what is meant by “fold change” in figures 1.b-g, 2.b-d, 7.b-d. It is unclear what the baseline is for fold comparison.

Response: Thank you for the question. In each case, the ‘fold-change’ is in comparison to the ‘No Target Control’ or NTC. This is the control wherein both the Pp and Pd activators are not present in the reaction. We have updated our captions to make this clearer.

R3.3.8 Could the authors include statistical comparisons with data presented in figure 2.f-h? Additionally, plotting these data on a Log scale could help significant differences be more apparent.

Response: Thank you for the suggestion. We have added statistical analysis to this figure.

Fig. 2 Split-activator detection of ssDNA, dsDNA, and RNA substrates by Cas12a: a. Schematic representation of Cas12a activated by combinations of ssDNA (red), dsDNA (orange), and RNA (blue) in the PAM proximal and PAM distal regions. b-d Heat maps representing the fold changes of *in vitro* trans-cleavage assay (n=3) with Cas12a orthologs for the combinatorial schemes seen in (a). e-h Comparison of

the WT crRNA and ENHANCE crRNA for *in vitro trans*-cleavage assay split activators. Note, ssDNA and ssRNA substrates were used as targets in the Pd region while dsDNA was supplied in the Pp. Reactions were incubated for 60 min at 37°C. Error bars represent SD (n=3). The reactions contained 25 nM of each truncated activator, 60 nM Cas12a, and 120 nM crRNA.

R3.3.9 Can the authors explain why a dashed line is included in figure 3.b-d at 10×10^5 ?

Response: Thank you for the question. We added this dashed line as a reference for comparing the data between the three charts. We found it difficult to visually identify the differences between LbCas12a and ErCas12a

R3.3.10 The sentence “To test these targets, we designed two crRNAs – a canonical WT as well as a SAHARA crRNA that only binds to the target at the Pd end while binding to a synthetic S12 at the Pp-end.” Suggests SAHARA crRNAs are different from WT crRNAs. Could the authors provide more details on how the sequences differ in the region that would bind the S12 duplex?

Response: Thank you for the question. As depicted in Fig. 3a, the spacer region of the WT crRNA binds completely to the target RNA. However, for the SAHARA crRNA, only the Pd end of the crRNA binds to the target, while the Pp end is designed to bind to a S12 DNA sequence that is separate and not a part of the RNA target.

To make the visualization clearer to the reviewer, we have reproduced the target sequence and the WT and S12 DNA sequence below:

Target RNA: 5' AGCACCCAGUCCGCCUGAG 3'

S12 DNA: 5' TTAATGCTAATCTAAAGCG 3'

WT crRNA: 5' CUCAGGGCGGACUGGGUGCU 3'

SAHARA crRNA: 5' GAUUAGCAUUAACUCAGGGCGGAC 3'

R3.3.11 In figure 4, the limit of detection is indicated by a dashed line in figure 4.c. It would be appropriate to include a similar line indicating limit of detection to figure 4.f

Response: Thank you for the suggestion, we have added a dashed line for Fig. 4f.

R3.3.12 In some figures, only the colors are used to symbolize the different Cas12a orthologs (Lb, As, and Er as Red, Green and Orange, respectively). The authors should indicate with text in each figure and in each figure legend which orthologs are used to generate the data in case the figures are printed in black and white and to accommodate color-blind individuals.

Response: Thank you for this important suggestion. We have added Cas12a ortholog labels to every figure.

R3.3.13 Change “Pp 10-nt nucleotides of the crRNA protospacers” to “Pp 10-nt nucleotides of the crRNA spacer”

Response: Thank you for the suggestion. We have made this change.

R3.3.14 The sentence “To study the effect of different metal ions on the activity of SAHARA, we tested a range of different metal cations (NH₄⁺, Rb⁺, Mg²⁺, Zn²⁺, Cu²⁺, Co²⁺, Ca²⁺, Ni²⁺, Mn²⁺, and Al³⁺) and discovered that most metal ions including severely inhibited the trans-cleavage activity of SAHARA (Fig. S9).” The word “metal” should be removed as NH₄⁺ is not a metal ion, and the word “including” should be removed to clarify the meaning of the sentence.

Response: Thanks for pointing that out. We made the necessary changes.

To study the effect of different metal ions on the activity of SAHARA, we tested a range of different cations (NH₄⁺, Rb⁺, Mg²⁺, Zn²⁺, Cu²⁺, Co²⁺, Ca²⁺, Ni²⁺, Mn²⁺, and Al³⁺) and discovered that most cations severely inhibited the trans-cleavage activity of SAHARA (Fig.S9).

REFERENCES

1. Long T. Nguyen, Nicolas C. Macaluso, & Piyush K. Jain. A Combinatorial Approach towards Adaptability of 22 Functional Cas12a Orthologs for Nucleic Acid Detection in Clinical Samples. *medRxiv* 2021.07.21.21260653 (2021) doi:10.1101/2021.07.21.21260653.
2. Nguyen, L. T., Smith, B. M. & Jain, P. K. Enhancement of trans-cleavage activity of Cas12a with engineered crRNA enables amplified nucleic acid detection. *Nat. Commun.* **11**, 4906 (2020).
3. Li, Q. *et al.* Synergistic Incorporation of Two ssDNA Activators Enhances the Trans-Cleavage of CRISPR/Cas12a. *Anal. Chem.* (2023) doi:10.1021/acs.analchem.3c00414.
4. Jinek Martin *et al.* A Programmable Dual-RNA-Guided DNA Endonuclease in Adaptive Bacterial Immunity. *Science* **337**, 816–821 (2012).
5. Zetsche, B. *et al.* Cpf1 is a single RNA-guided endonuclease of a class 2 CRISPR-Cas system. *Cell* **163**, 759–771 (2015).
6. Swarts, D. C., van der Oost, J. & Jinek, M. Structural Basis for Guide RNA Processing and Seed-Dependent DNA Targeting by CRISPR-Cas12a. *Mol. Cell* **66**, 221-233.e4 (2017).
7. Chen, J. S. *et al.* CRISPR-Cas12a target binding unleashes indiscriminate single-stranded DNase activity. *Science* **360**, 436–439 (2018).
8. Jeon, Y. *et al.* Direct observation of DNA target searching and cleavage by CRISPR-Cas12a. *Nat. Commun.* **9**, 2777 (2018).
9. Swarts, D. C. & Jinek, M. Mechanistic Insights into the cis- and trans-Acting DNase Activities of Cas12a. *Mol. Cell* **73**, 589-600.e4 (2019).
10. Stella, S. *et al.* Conformational Activation Promotes CRISPR-Cas12a Catalysis and Resetting of the Endonuclease Activity. *Cell* **175**, 1856-1871.e21 (2018).
11. Singh, D. *et al.* Real-time observation of DNA target interrogation and product release by the RNA-guided endonuclease CRISPR Cpf1 (Cas12a). *Proc. Natl. Acad. Sci.* **115**, 5444–5449 (2018).

12. Dmytrenko, O. *et al.* Cas12a2 elicits abortive infection through RNA-triggered destruction of dsDNA. *Nature* **613**, 588–594 (2023).
13. Wierson, W. A. *et al.* Expanding the CRISPR Toolbox with ErCas12a in Zebrafish and Human Cells. *CRISPR J.* **2**, 417–433 (2019).
14. Magnusson, J. P., Rios, A. R., Wu, L. & Qi, L. S. Enhanced Cas12a multi-gene regulation using a CRISPR array separator. *eLife* **10**, e66406 (2021).
15. Creutzburg, S. C. A. *et al.* Good guide, bad guide: spacer sequence-dependent cleavage efficiency of Cas12a. *Nucleic Acids Res.* **48**, 3228–3243 (2020).

Reviewers' Comments:

Reviewer #1:

Remarks to the Author:

In this resubmission, the authors have addressed all previous concerns. No further questions from this reviewer.

Reviewer #2:

Remarks to the Author:

The authors elaborately responded to my and other reviewer's comments and have added additional data that strengthens the conclusions of the manuscript. I have one last very minor comment I would like the authors to address.

R2.6.29 Figure legend 1b-d: "Fold change at t=60 minutes of in vitro trans-cleavage assay..."
Please revise (what does fold change of an assay mean, fold change in what, compared to what?)

Response: Thank you for the feedback, we have added a definition for the fold change as "fold change normalized to NTC" in the figure caption.

The authors added the requested information on how the fold change was normalized, but did not add what the was measured to calculate the fold change, fluorescence?

Reviewer #3:

Remarks to the Author:

All of the initial concerns I had with the manuscript have been resolved. I recommend that this interesting work be published.

Reviewer #1 (Remarks to the Author):

In this resubmission, the authors have addressed all previous concerns. No further questions from this reviewer.

Response: Thank you for your positive feedback and the time you've dedicated to reviewing our manuscript. We are glad to hear that our responses were satisfactory.

Reviewer #2 (Remarks to the Author):

The authors elaborately responded to my and other reviewer's comments and have added additional data that strengthens the conclusions of the manuscript. I have one last very minor comment I would like the authors to address.

R2.6.29 Figure legend 1b-d: “Fold change at t=60 minutes of *in vitro* trans-cleavage assay...” Please revise (what does fold change of an assay mean, fold change in what, compared to what?)

Response: Thank you for the feedback, we have added a definition for the fold change as “fold change normalized to NTC” in the figure caption.

The authors added the requested information on how the fold change was normalized, but did not add what the was measured to calculate the fold change, fluorescence?

Response: Thank you for your positive feedback and the time you've dedicated to reviewing our manuscript. We are glad to hear that our responses were satisfactory.

Regarding the last minor comment, the fold change of the fluorescence intensity was calculated. We have made the following changes to the figure caption.

Fig. 1: Cas12a orthologs tolerate short ssDNA activators (6-12 nt) when added in combination a. Schematic representation of a crRNA-Cas12a complex performing *trans*-cleavage of ssDNA reporters following the recognition of two split-activators. **b-d.** Fold change of fluorescence intensity normalized to No Target Control (NTC) at t=60 minutes of *in vitro* trans-cleavage assay with Cas12a orthologs (red = LbCas12a, green = AsCas12a, orange = ErCas12a) activated by individual truncated ssDNA activators of length 6-20 nt. Statistical comparisons are made for a combination of both Pp and Pd data sets between the different lengths of the targets...

Reviewer #3 (Remarks to the Author):

All of the initial concerns I had with the manuscript have been resolved. I recommend that this interesting work be published.

Response: Thank you for your positive feedback and the time you've dedicated to reviewing our manuscript. We are glad to hear that our responses were satisfactory.